# Sharp contrasts in observed and modeled crevasse patterns at Greenland's marine terminating glaciers

Ellyn M. Enderlin[1], Timothy C. Bartholomaus[2]

[1]Department of Geosciences, Boise State University, Boise, Idaho, 83725, USA
[2]Department of Geological Sciences, University of Idaho, Moscow, Idaho, 83844, USA

*Correspondence to*: Ellyn M. Enderlin (ellynenderlin@boisestate.edu)

**Abstract.** Crevasses are affected by and affect both stresses and surface mass balance of glaciers. These effects are brought on through potentially important controls on meltwater routing, glacier viscosity, and iceberg calving, yet there are few direct observations of crevasse sizes and locations to inform our understanding of these interactions. Here we extract depth estimates for the visible portion of crevasses from high-resolution surface elevation observations for 52,644 crevasses from 19 Greenland glaciers. We then compare our observed depths with those calculated using two popular models that assume crevasse depths are functions of local stresses: the Nye and linear elastic fracture mechanics (LEFM) formulations. When informed by the observed crevasse depths, the LEFM formulation produces kilometer-scale variations in crevasse depth in decent agreement with observations. However, neither formulation accurately captures smaller scale variations in the observed crevasse depths. Furthermore, we find that along-flow patterns in crevasse depths are unrelated to along-flow patterns in strain rates (and therefore stresses), but that cumulative strain rate is moderately more predictive of crevasse depths at the majority of glaciers. Based on these results, acknowledging the limitations of lidar-derived crevasse depth estimates, we recommend additional crevasse analyses that leverage in situ observations and a variety of remote sensing datasets to thoroughly test the Nye and LEFM models. These analyses must be performed in extensional and compressional regions to better understand the influence of advective processes on crevasse geometry, so that projections of terminus position change or meltwater routing that rely on crevasses can adequately map patterns in crevasse geometry for varying strain fields.

## 1 Introduction

The geometry and concentration of crevasses are both affected by and affect the stress state and surface mass balance of glaciers, ice shelves, and ice sheets (Colgan et al., 2016). Changes in crevasse geometry and concentration can arise as the result of long-term or rapid changes in stress state, serving as a valuable tool to infer the onset of kinematic change (Colgan et al., 2011; Trantow and Herzfeld, 2018). These changes can also influence the stress state. For example, changes in crevassing within lateral shear margins of Antarctic ice streams have the potential to dramatically alter the ability of ice streams to buttress flow from the interior, in turn exerting an important control on ice sheet stability (Borstad et al., 2016; Reese et al., 2018). The impoundment of surface meltwater runoff in crevasses is expected to promote crevasse penetration and assist in the penetration of meltwater to the glacier bed, thereby influencing the englacial and basal stress states (van der Veen, 1998; Stevens et al., 2015; Poinar et al., 2017). Crevasses also increase surface roughness, altering the incidence angle of solar radiation and

turbulent energy fluxes, which in turn influence surface melt production (Pfeffer and Bretherton, 1987; Andreas, 2002; Hock, 2005; Cathles et al., 2011; Colgan et al., 2016).

These interactions between crevasses, stresses, and surface mass balance make crevasses particularly important components of terrestrial ice, particularly near the termini of marine-terminating glaciers and ice streams draining Greenland and Antarctica. In Antarctica, observations and models indicate that the ice shelves fringing the continent are highly vulnerable to widespread crevasse hydrofracture in a warming climate (Pollard et al., 2015; Rott et al., 1996; Scambos et al., 2000, 2009). The influence of crevasses, and changes in crevassing over time due to atmospheric warming, are less clear for Arctic marine-
terminating glaciers. Despite the abundance of crevasses throughout the marginal zone of the Greenland ice sheet, there are few observations of crevasse depths at Greenland's glaciers. However, the coincident increase in surface meltwater runoff and widespread retreat of glacier termini across Greenland (Carr et al., 2017; Howat and Eddy, 2011; Moon and Joughin, 2008) suggests that hydrofracture may exert a first-order control on calving (Benn et al., 2007a; Cook et al., 2014).

Since calving involves the mechanical detachment of ice from a glacier terminus, it has been assumed that calving occurs when and where surface crevasses penetrate the full ice thickness (Benn et al., 2007a). The penetration depth of crevasses has been traditionally modeled using either the linear elastics fracture mechanics (LEFM) or Nye formulations, which assume that crevasse depth is dictated by local longitudinal stresses. The penetration of fractures and crevasse tips beyond the open portion at the glacier surface (Colgan et al., 2016), means that the exact validation of either model is not possible at whole-glacier or
ice sheet scales using currently available methods. However, under the assumption that observable crevasse shapes and depths are in some way related to the full depths of fractures, Mottram and Benn (2009) compared 44 open crevasse depths, measured in the field, with modeled depths at Breiðamerkurjökull, Iceland. They found that measured crevasse depths were not especially predictive of either the LEFM- or the Nye-predicted crevasse depths ($R^2 \leq 0.2$ for all models), but that the LEFM formulation is more accurate than the Nye formulation when a priori information on crevasse geometries are available to constrain $F(\lambda)$.

Despite potential model shortcomings, and the potential over-estimation of crevasse depths by the Nye formulation found at Breiðamerkurjökull, Iceland (Mottram and Benn, 2009), the Nye formulation has been implemented in a number of numerical ice flow models as the terminus boundary condition (Cook et al., 2014; Nick et al., 2013; Vieli and Nick, 2011). The Nye formulation has been used in lieu of the LEFM formation because observations of dense crevasse fields near glacier termini
suggest considerable stress blunting occurs (limiting penetration depths) and the a priori information on crevasse geometries needed to tune the LEFM formulation have been lacking. Using Eq. 2, the ice viscosity or water depth in crevasses can conceivably be tuned to drive changes in crevasse depth, so that the modeled terminus position change matches observations. The high sensitivity of simulated terminus positions to changes in water depth (Cook et al., 2012, 2014; Otero et al., 2017), however, lead us to question the appropriateness of this model. Because model projections of dynamic mass loss may well be

in error if driven by an inaccurate calving parameterization, increased confidence in dynamic mass loss projections drawing on these calving parameterizations requires validation of modeled crevasse depths.

To improve our understanding of crevasse occurrence and enable model validation, we construct here the first extensive record of observable crevasse depths for Greenland's fast-flowing outlet glaciers using airborne lidar and high-resolution digital

elevation models from 2011-2018. Although ice fracturing likely extends below the open portion of crevasses mapped using this method, we compare our open crevasse depth observations with crevasse depths modeled using satellite-derived strain rate fields similar to Mottram and Benn (2009).  Furthermore, we examine the likelihood that spatio-temporal variations in crevasse depth can explain observed variations in terminus position change and associated dynamic mass loss for Greenland's marine-terminating outlet glaciers. Although we focus on Greenland, our assessment of 19 glaciers spanning a wide range of

geometries, climate regimes, and dynamic histories (Fig. 1a) increases the likelihood that our results are broadly applicable to glaciers throughout the Arctic.

## 2 Methods

### 2.1 Observed Crevasse Depths

Surface crevasses exist as fractures in ice that extend from glacier surfaces, generally as some sort of visible opening within a glacier, and continue down to the bottom of a fracture, to unbroken ice. There is currently no technique to measure the full depth of fracturing that extends beneath the open portion of surface crevasses. Thus, our analysis begins with measurements of surface topography from which we estimate the depth of the open portion of crevasses, i.e., that portion of a crevasse that can be observed with visible light. This observed crevasse depth represents a minimum estimate of the true crevasse depth.


We construct time series of observed crevasse depths from flow-following lidar swaths acquired by NASA Operation IceBridge (OIB) and 2 m-resolution digital elevation models (DEMs) using a semi-automated approach that identifies crevasses from local elevation minima (Figs. 1b-e). We use lidar observations from the OIB ATM (Advanced Topographic Mapper), which has a vertical precision of better than 1cm and spatial sampling of one pulse every ~10 m$^2$ within its conical

swath (https://nsidc.org/data/ilatm1b). Repeat April/May flow-following observations are available for all our study sites during the 2013-2018 period. Elevations were also extracted from 2 m-resolution DEMs produced by the Polar Geospatial Center as part of the ArcticDEM program. The WorldView DEMs are less precise (3m vertical uncertainty (Noh and Howat, 2015)) but provided estimates of elevation throughout the 2011-2018 melt seasons. We used an average of ~4 lidar swaths and ~16 DEMs per glacier for our analysis.


Lidar swaths were overlain on cloud-free summer Landsat 8 images and swath centerlines were manually traced to the inland extents of visible crevassing. Using a moving window approach, shifted at ~1 m increments along the swath centerlines, we linearly interpolated the nearest elevation data, then identified crevasses using a filtering process described below and illustrated for Kong Oscar Gletsjer in Fig. 1. To identify crevasses, centerline elevations were first detrended over the ~500 m-wide moving window (Fig. 1b inset), then the local elevation minimum and maximum were extracted from each of three smaller windows centered on the detrended profile (Fig. 1b, gray shading). The process was repeated over the full profile length, resulting in the identification of local lows and highs for each elevation profile. The minimum elevation was identified from each grouping of contiguous low points, a similar procedure was used to define high points, and the remaining points were discarded (Fig. 1c). For each minimum, the closest neighboring down- and up-glacier maxima were used to define longitudinal crevasse widths (Fig. 1d). Potential collapsed seracs at the bottom of crevasses and small surface irregularities less than the vertical uncertainty of the DEMs were discarded.

The appropriate lengths for the detrending window and search windows to identify the local minima were determined through a comparison of manual and automated crevasse depth distributions (i.e., depths and their locations) from the most complete lidar profile for each glacier. Six detrending window sizes and two sets of search window sizes were tested, for a total of 12 test combinations, as outlined in Table S1. The range of possible detrending window sizes was constrained by the requirements that the window (1) spanned the largest crevasses (~200 m in width at Helheim Glacier) and (2) did not exceed the maximum half-wavelength of large-scale oscillations in surface elevation evident along the profiles (~800-1500m). For the search windows, we tested sizes that minimally spanned the maximum observed half-width of crevasses, but fully spanned the majority of crevasses: the median width ± 1.48 MAD (median of absolute deviation) of the 3264 manually-identified crevasses, which is analogous to the mean ± the standard deviation for normally-distributed data, was 19.2 ± 10.2 m and the maximum width was 184 m. The optimal window combination used for automated crevasse identification was the window combination that yielded the smallest number of falsely-identified crevasses (both false positives and false negatives) and the smallest depth misfit relative to the manually-extracted dataset. Optimal window sizes were glacier-dependent, with a size of 350 m identified for 9 glaciers, 500 m for 2 glaciers, 550 m for three glaciers, 650 m for one glacier, and 800 m for four glaciers. The smaller search window sizes were considered optimal for all study sites except Helheim Glacier, which had the widest crevasses. For these optimal window sizes, the median false negative rate was 1.2% and the median false positive rate was 38.5% across all glaciers. In other words, the automated algorithm missed ~1% of manually-identified crevasses and identified ~38% more crevasses than the manual interpreter.

Given the off-nadir scan angle of the OIB lidar and the stereo imagery used to construct the DEMs, it is highly unlikely that the elevation observations penetrate to the bottom of the open portion of all crevasses. As in Liu et al. (2014), who used ICESat data to estimate crevasse depths across the Amery Ice Shelf, we found that most crevasses followed V-shaped geometries, although some crevasses had flattened floors (Fig. 1). Based on the commonality of V-shaped crevasses in our elevation

transects, we assumed that crevasses initially formed with planar geometries extending to some ultimate fracture depth, and that further extensional strain opens these planar factures into V shapes. As stated above, further fracturing below the bottom of the V is possible, although we would then expect it to also open under the same extensional strains that opened the V shapes above. Apparent deviations from an idealized V-shaped geometry are likely due to serac toppling, over-printing of new crevasses on previously damaged ice (Colgan et al., 2016), the presence of impounded meltwater, ice debris in the crevasse,

or occlusion of the lidar signal by the walls of the crevasse due to the off-nadir pointing geometry of the airborne lidar. Assuming that open crevasses are truly V-shaped, we estimated the true depth of the open portion of each crevasse by linearly projecting both crevasse walls to depth and identified their extrapolated point of intersection (Fig. 1e). For each elevation minimum and closest neighboring down- and up-glacier maxima, the crevasse walls were identified as contiguous regions with slopes within the typical range observed for manually-identified V-shaped crevasses in the window-calibration elevation

profiles. Since there is no physical reason why the crevasse wall surface slopes should be normally distributed, we used the median ± 1.48 MAD to characterize the typical range. For irregularly-shaped crevasses and for crevasses located where the rough glacier surface resulted in local elevation maxima several meters to tens of meters from the crevasse edge, this approach retracted the crevasse wall extents to correspond with slope breaks. If wall slopes were entirely outside of the typical range, there was no effect on the crevasse extents. We refer to the average elevation difference between the top and extrapolated

bottom of crevasses as the observed crevasse depths. Since we extrapolate crevasse depths using the wall slopes, we do not attempt to filter our data to exclude crevasses where the lidar may not have reached the bottom of the open portion of the crevasse. Again, as our observed crevasse depths only capture the open portion of crevasses, these represent minimum estimates of ice fracture depths extending from the surface.

We estimated uncertainties associated with (1) spatial resolution of the satellite-derived DEMs through comparison of same-day ATM profiles, (2) the automated approach for crevasse identification through comparisons with depths from manually-identified crevasses, and (3) crevasse depth extrapolation through comparisons between observed and extrapolated depths for V-shaped crevasses. All values presented are the median ± 1.48 MAD unless otherwise stated.

Although the precision of the lidar elevations is better than 1cm, the discrete sampling of the lidar may not be coincident with the location of the true crevasse bottom. Uncertainties associated with the lidar spatial sampling were quantified through a comparison of crevasse depths extracted from same-day up- and down-glacier swaths. The difference in observed crevasse depths between repeat swaths was -0.35 ± 2.5 m. We attribute the non-zero mean depth difference to differences in the imaging angle of the lidar between repeat flights and its influence on the calculated crevasse wall slopes. Uncertainties associated with

the inclusion of the lower resolution WorldView DEM-derived depths were estimated through a comparison of same day lidar- and DEM-derived crevasse depths. We found that the DEM-derived depths were 1.0 m less than the lidar-derived depths, with a MAD of 2.5 m. A comparison of high-resolution and 2 m-resolution lidar-derived crevasse depths indicated the decrease in horizontal resolution of the DEMs accounted for ~1/3 of the DEM-derived depth bias. Since the potential biases were within

the uncertainties in the datasets, we do not discuss them further. The lidar-derived depth uncertainty and the MAD from the lidar-DEM depth comparison were summed in quadrature to obtain a DEM-derived depth uncertainty of 3.0 m.

Uncertainties associated with automated crevasse depth estimation were quantified through a comparison of manually- and automatically-extracted crevasse depths. Automation uncertainties were minimized through the use of manual calibration datasets. Typical uncertainties introduced by the use of our automated approach were -0.3 ± 0.6 m, indicating that the automated approach slightly over-estimated observed crevasse depths due to differences in the manual versus automated identification of crevasse wall limits.

Our assumption of V-shapes for the open portion of crevasses was supported by observations of abundant V-shaped crevasse openings in every elevation profile examined here. For the V-shaped crevasse openings identified in the calibration profiles, the difference between the observed and extrapolated depths was <0.1 m on average. Examples of V-shaped crevasse openings can be found in Fig. 1 and scatterplots comparing observed and extrapolated depths for V-shaped and irregularly-shaped crevasses are shown in Fig. S1.

Overall, we estimate lidar-derived and DEM-derived depth uncertainties of 2.5 m and 4.4 m, respectively, with the tendency toward slight under-estimation of observed crevasse depths (1.0 m bias) when using DEMs. Automation results in a slight over-estimation (0.3 m) of observed crevasse depths due to differences in the manual and automated crevasse wall extents. The difference between observed and extrapolated crevasse depths for V-shaped crevasses is <0.1 m, indicating an excellent linear crevasse wall approximation and inconsequential bias associated with extrapolated depths (Fig. S1). We are unable to assess the extent to which micro-fractures may extend beyond the depths of these observed crevasses. Additionally, to our knowledge, no other dataset exists that can validate our remotely-sensed estimates of open crevasse depths or the relationship between open crevasse depths and full depth of fractures.

## 2.2 Modeled Crevasse Depths

Crevasse depths were modeled using the Nye and Linear Elastic Fracture Mechanics (LEFM) formulations. For an individual crevasse, the LEFM formulation models the stress concentration at the crevasse tip as

$$K_I = F(\lambda)R_{xx}\sqrt{\pi d_{LEFM}}, \tag{1}$$

where $d_{LEFM}$ is the modeled crevasse depth, $R_{xx}$ is the full stress minus the lithostatic stress (estimated using strain rates), and $F(\lambda)$ is a geometric parameter that accounts for the non-linear increase in the stress intensity factor as a crevasse penetrates deeper into the glacier and the ratio of the crevasse depth to ice thickness, $\lambda$, increases (van der Veen, 1998). For this formulation, the crevasse will penetrate to the maximum depth where the stress concentration is sufficient to overcome the

fracture toughness of the ice, $K_{IC}$ (van der Veen, 1998). We rearranged Eqn. 1 to model crevasse depths using the LEFM formulation as

$$d_{LEFM} = \frac{1}{\pi}\left(\frac{K_{IC}}{F(\lambda)R_{xx}}\right)^2,$$ (2)

As the spacing between crevasses decreases, stress concentration is progressively blunted. For closely-spaced crevasses, concentration of stresses at crevasse tips can be ignored and surface crevasse depths can be estimated using the Nye formulation (Nye, 1957), such that

$$d_{Nye} = \frac{2}{\rho_i g}\left(\frac{\dot{\varepsilon}_{xx} - \dot{\varepsilon}_{crit}}{A}\right)^{1/3} + \frac{\rho_w}{\rho_i}h_w,$$ (3)

where $\rho_i$ and $\rho_w$ are the densities of ice (917 kg m$^{-3}$) and water (1000 kg m$^{-3}$), $g$ is gravitational acceleration (9.81 m s$^{-2}$), $\dot{\varepsilon}_{xx}$ is the longitudinal strain rate (yr$^{-1}$), $\dot{\varepsilon}_{crit}$ is the critical strain rate threshold for crevasse formation (yr$^{-1}$), $A$ is the creep parameter describing ice viscosity (Pa$^{-3}$ yr$^{-1}$), and $h_w$ is the depth of water in crevasses. The critical strain rate, creep parameter, and crevasse water depth were estimated for each glacier as described below.

The LEFM- and Nye-modeled crevasse depths represent the full depth of fractures extending from the surface. Here we primarily focused on the Nye formulation given its more widespread use in calving parameterizations. For both formulations, crevasses were only expected under tension, with the deepest crevasses in locations with the highest longitudinal stresses and most viscous (i.e., colder and/or less damaged) ice. Neither formulation accounted for the inheritance of damaged ice from upstream, meaning the crevasse depths were estimated as functions of the local, instantaneous, longitudinal stress without consideration of crevasse advection.

To solve for modeled crevasse depths, strain rates were computed from NASA Making Earth System Data Records for Use in Research Environments (MEaSUREs) Interferometric Synthetic Aperture Radar velocities (https://nsidc.org/data/NSIDC-0481/versions/1). The temporal coverage of these approximately bi-weekly velocity fields varied widely between glaciers, with an average of 66 velocity maps per glacier and a maximum of 282 maps for Jakobshavn from 2011-2018. Spatial gradients in velocity were used to compute strain rates in the native (polar stereographic) coordinate system, which were then rotated into flow-following coordinates. The creep parameter ($A$) is dependent on a number of variables, including ice temperature, crystal fabric development, and damage, but is poorly constrained by observations. Here, we approximated temperature-dependent spatial variations in the creep parameter as a linear function of latitude (Nick et al., 2013). Longitudinal strain rates were calculated over the full velocity domain then linearly interpolated to the swath centerlines.

For our initial estimates using the Nye model, what we term the 'minimal' model, we followed the approach of Mottram and Benn (2009) and assumed crevasses formed everywhere under tension (i.e., no critical strain rate threshold) and there was no water in crevasses (likely the case for spring OIB data). To improve model performance, we also tested several more complex

versions of the model. We first estimated the critical strain rate for crevasse formation at each glacier as the maximum strain rate inland of the most up-glacier crevasse observation. To explore the potential contribution of the ice viscosity parameterization to the observed-modeled depth discrepancy, we assumed that the observed crevasse depths are accurate and tuned the ice viscosity parameter to minimize the misfit between observed and modeled crevasse depths. Similar to Borstad et al. (2016), we included a deformation enhancement factor, $D$, in these calculations as

$$d_{observed} = (1 - D) \left[ \frac{2}{\rho_i g} \left( \frac{\dot{\varepsilon}_{xx}}{A} \right)^{1/3} \right]. \tag{4}$$

Substituting our initial modeled crevasse depths (i.e., Eq. (3) with $\dot{\varepsilon}_{crit} = 0$ and $h_w = 0$) in for the RHS term in brackets and rearranging to solve for the deformation enhancement factor, we obtained

$$D = \frac{d_{Nye} - d_{observed}}{d_{Nye}}. \tag{5}$$

Although similar to damage in Borstad et al. (2016), our deformation enhancement factor is likely a function of spatial variations in damage, ice temperature, and crystal fabric. A unique deformation factor can be identified at each crevasse location using Eq. (5). However, such detailed tuning is neither physically motivated nor practical for models since numerous processes can contribute to the misfit between observed and modeled crevasse depths, so we binned the data along-flow then parameterized deformation enhancement as a linear function of distance from the terminus using the binned data (Fig. S2). The deformation enhancement factors for the deepest crevasses over each 300 m bin, spanning from the terminus to the inland-most crevasse observation, were used in our parameterizations. Finally, we also used the inland-most deformation enhancement value to solve for modeled crevasse depths under the assumption of spatially uniform ice viscosity, then estimated impounded water depths from the misfit between the observed and modeled crevasse depths. Again, we sought a simple parameterization appropriate for use in numerical ice flow models: assuming that water depth varies with meltwater generation, we parameterized impounded water depth as linear function of surface elevation for each glacier (Fig. S3). For the damage and impounded water depth parameterizations, bin size did not influence along-flow patterns discussed below.

Although numerical ice flow models have relied on the Nye formulation to model crevasse depths, the previously-observed over-estimation of crevasse depths by the Nye formulation relative to both observations and the LEFM formulation (Mottram and Benn, 2009) suggests there may be large differences in accuracy of the Nye and LEFM formulations. We used Eqn. (2) with $K_{IC}$ =50kPa m$^{1/2}$ as our best estimate for the fracture toughness of ice and constrained uncertainty in this term using minimum and maximum values of 10kPa m$^{1/2}$ and 100kPa m$^{1/2}$, respectively. Following convention, the longitudinal stress, $\sigma_{xx}$, was calculated from the measured strain rate tensors using

$$\sigma_{xx} = A^{-1/n} \dot{\varepsilon}_e^{(1-n)/n} \dot{\varepsilon}_{xx}, \tag{5}$$

where $\dot{\varepsilon}_e$ and $\dot{\varepsilon}_{xx}$ are the second invariant of the strain rate tensor (i.e., effective strain rate, assuming negligible vertical shear) and the longitudinal strain rate in the direction of ice flow, respectively, and n=3. The lithostatic stress was subtracted from $\sigma_{xx}$ to estimate the longitudinal resistive stress, $R_{xx}$. Longitudinal resistive stress was calculated over the full velocity domain,

averaged in time, then linearly interpolated to the swath centerlines. Since Eqn. (5) cannot account for stress relief caused by crevasse formation, these longitudinal resistive stress estimates should be considered upper estimates of the stress state of the ice. As in Mottram and Benn (2009), $\lambda$ was calculated at each observed crevasse location using the observed crevasse depth to ice thickness ratio (van den Broeke et al. 1998); Eqn. (6)) in an effort to utilize our a priori information on crevasse depth to minimize the misfit between observed and modeled crevasse depths.

## 3 Results

### 3.1 Observed Crevasse Depths

We identified a total of 52644 open crevasses in 381 elevation profiles among the 19 study glaciers (Enderlin, 2019). Broadly, we see no clear, consistent patterns in either the crevasse density or depths from the interior towards the terminus across all glaciers in our analysis. The distributions of observed (i.e., open) crevasse depths are shown in Fig. 2 and observed depth
profiles are shown in Fig. S4. We present statistics pertaining to observed crevasse depth and concentration, i.e., number of crevasses per kilometer, within 5 km of glacier termini in Table 1. Of all observed crevasses, the median open depth was 6.2 m and median concentration was 17.2 open crevasses per kilometer (one crevasse every 58 m). The crevasse concentrations span a fairly narrow range of values, with ~75% of crevasse concentrations between 15.0-19.7 crevasses km$^{-1}$, despite a wide range of glacier thicknesses and strain rates. The two least crevassed glaciers (concentrations less than 10 km$^{-1}$) have floating
tongues and occur in the coldest, high latitude regions. The maximum observed depth of 64.9 m occurred at steep, fast-flowing Helheim Gletsjer. Helheim also had the deepest median observed crevasse depth of 10.2 m. While some glaciers have more and deeper crevasses near the terminus than inland, this pattern is clearly not universal, and in many instances, open crevasse depths decreased over the last several km of the terminus region (Figs. 2, 3, S4).

Although the crevasse size distributions are dominated by a large number of relatively shallow (i.e., <10 m-deep) crevasses, we are primarily interested in the deepest crevasses, which are the most likely to penetrate the full glacier thickness and therefore play an important role in iceberg calving and meltwater routing to the glacier bed. To isolate the deepest crevasses from the observations, we identified the maximum crevasse depth at 150 m-increments along flow so that the along-flow variations in crevasse depth had the same spatial resolution as the velocity data used to compute strain rates. To determine
whether along-flow variations in maximum observed crevasse depth can be explained by either local variations in local longitudinal strain rates or strain (i.e., time-integrated longitudinal strain rate), we normalized the observed crevasse depth, strain rate, and strain data to facilitate direct comparison of their along-flow patterns. Data were linearly normalized such that the observed values span from zero to one. The normalized profiles in Fig. 3 suggest that along-flow variations in maximum crevasse depth cannot be simply explained as a function of variations in either local strain rate or strain across all glaciers,
although kilometer-scale variations in maximum crevasse depth appear to follow patterns in strain at approximately half of the glaciers (Figures S5-S7).

## 3.2 Crevasse Depth Comparison

Given that our observed crevasse depths are limited to the open portion of crevasses and the modeled crevasse depths represent the full depth of fractures extending from the surface, we expect that the observed depths will be less than modeled depths in regions of longitudinal extension (i.e., where the models predict crevassing). We indeed find this pattern (Figs. 4 and S8-S25). However, a comparison of spatio-temporal variations in the difference between observed and modeled crevasse depths can yield insights into controls on crevassing. As demonstrated for Inngia Isbræ in Fig. 4 (other glaciers in the supplement), observed crevasse depths were generally less than predicted using the minimal Nye model (points in white region) but the model under-estimates crevasse depths or fails to predict them entirely in some locations (points in gray region). Where crevasses were observed but strain rates were negative, crevasses were missed by the model and data fall along the x-axis. Although the maximum misfit and occurrence of missed crevasses decreased at longer spatial scales due to smoothing of the strain rate estimates, discrepancies between observed and modeled depths on the order of tens of meters were observed at all spatial scales. We find no correlation between the modeled and observed, minimum crevasse depths.

The comparisons of observed and modeled crevasse depths in Fig. 4 and Figs. S8-S25 also suggest that crevasse depths remained relatively stable at all study glaciers over the 2011-2018 period. Inngia Isbræ exhibited the largest dynamic change among our study glaciers – the glacier retreated by ~4 km and thinned by ~100 m near the terminus (Fig. 4a) and flow accelerated by ~500 m/yr near the terminus (not shown) from 2012-2017 – yet modeled crevasse depths do not significantly differ over time, and nearly all observed crevasse depths remain < 30 m throughout the observation record (Fig. 4b). The stable and consistent nature of the kilometers-scale fluctuations in crevasse depth are also visible for each glacier in Fig. S4. Uncertainties are not included in Fig. S4, but a large portion of the variations in crevasse depth are within the estimated uncertainty of ~3 m for the observed depths.

We illustrate along-flow variations in the discrepancy between modeled and observed crevasse depths at four study sites – Kong Oscar (northwest Greenland), Inngia (west), Daugaard-Jensen (east), and Heimdal (southeast) – in Fig. 5. For each panel, we represent temporal variability in modeled depths (driven by strain rate changes) in a minimal model (Fig. 5, orange shading, see Methods), but finding no clear pattern in the temporal variability, only identify modeled depths computed from the median speed profile for the remainder of our analysis (Fig. 5, colored lines). The complete set of plots, arranged geographically, are included in the supplemental material (Fig. S26).

Modifications of the minimal Nye model, including model variations with a threshold strain rate, different viscosities, and impounded water depth parameters all modified the spatial patterns in the difference between the observations and models. For example, because ice has tensile strength and crevasses will not form where the strain rates (and therefore tensile stresses)

do not exceed an appropriate tensile strength-dependent threshold, we added a critical strain rate for crevasse formation into the Nye model. We found that the addition of an observation-based non-zero critical strain rate increased the extent of the modeled no-crevasse regions, resulting in kilometers-scale fluctuations in crevasse depth that contrast with the more gradual observed variations in crevasse depth (Fig. 5; red lines).

Along-flow variations in ice viscosity associated with strain-induced variations in crystal fabric or temperature, cryohydrologic warming, or even the presence of crevasses themselves may also contribute to differences between the modeled and observed crevasse depths. Inclusion of a deformation enhancement parameterization that varied linearly along flow (Fig. S2) reduced the magnitude of fluctuations in modeled crevasse depths so that the modeled and maximum observed crevasse depths were in better agreement (Figs. 5, S26; green lines). However, despite the expected along-flow increase in the deformation

enhancement factor with damage, strain heating, etc., minimization of modeled/observed crevasse depth misfits (Eq. 4) required an along-flow *decrease* in deformation enhancement for approximately half of the glaciers (Fig. S2).

Increasing crevasse water depths, potentially associated with increasing melt at low elevations, represent another potentially important process that can be parameterized in the Nye model. We used the inland-most deformation enhancement factor and

tuned impounded water depths to minimize the observed-modeled depth misfit. Water depths necessary for this minimization varied from 0 - 3.2 m for Zachariae Isstrøm up to as great as 32.7 m for Kong Oscar Gletsjer (Table 1). Modeled crevasse depths obtained using parameterized water depths are shown in Figs. 5 and S24 (blue lines). As with the deformation enhancement factor, we found inconsistent, positive and negative trends in crevasse water depth with along-flow variations in surface elevation. Only approximately half of the glaciers displayed patterns of increasing water depth with decreasing surface

elevation, as expected, while the remaining half of glaciers required either decreasing or no change in estimated water depths at the low-elevation, near-terminus regions (Fig. S3). Inclusion of a simple parameterization that scaled crevasse water depth as a linear function of elevation effectively smoothed the modeled crevasse depths so that they were better aligned the kilometers-scale patterns in observed crevasse depths (Fig. S26) but could not explain the smaller-scale oscillations in crevasse depth that we observed.


The LEFM model predicted crevasse depths of similar magnitude, and with comparable spatial patterns, as the damaged and hydrofracture-enhanced Nye models. Excluding regions of longitudinal compression, where both the LEFM and Nye formulations fail to predict crevassing (several tens of percent of glacier profiles), the median modeled depth for the minimal, $\dot{\varepsilon}_{crit}$ >0, damaged, and hydrofracture-enhanced versions of the Nye formulation exceeded the maximum observed crevasse

depths by an average of 29 m, 16 m, 0 m, and 2 m respectively. On average, LEFM depths are ~1 m deeper than the maximum observed depths under extension. However, like the Nye formulation, the LEFM model fails to reproduce realistic along-flow variations in crevasse depth for most glaciers. Figures 5 and S24 show the maximum LEFM crevasse depths averaged over 300 m bins (purple). The potential impact of uncertainty in fracture toughness is shown with purple shading, however, these

impacts are not visible at the scale of each panel and are obscured by the profiles for the intermediate fracture toughness value
($K_{IC}$ =50kPa m$^{1/2}$).

## 4 Discussion and Conclusions

Using the first spatially and temporally extensive record of surface crevasse depths for Greenland's fast-flowing marine-terminating glaciers, we find that there are typically >10 crevasses per kilometer but that the majority of the open portion of
crevasses are <10 m in depth. Given the skewed distributions of crevasse depths in Fig. 2, the inclusion of crevasses smaller than our detection threshold of 3 m-depth would likely increase the concentration and decrease the median depths relative to those reported in Table 1. Crevasse depths are highly variable along flow, with pronounced changes in the shapes of the crevasse depth distributions and maximum crevasse depths evident at most glaciers (Figs. 2, 3). Accumulated strain is an inconsistent predictor of large-scale variations in maximum crevasse depth, which follows strain at approximately half of our
study sites (Fig. 3, S6). Small-scale patterns in the observed crevasse depth cannot easily be explained by variations in local longitudinal strain rate, strain, or stress.

The discrepancy between modeled and observed crevasse depths, particularly the presence of crevasses up to 10s of meters deep in compressional zones where there are no modeled crevasses, is problematic for numerical ice flow models that rely on
spatio-temporal variations in crevasse depth to prescribe the terminus position. If calving is the result of open crevasse penetration to the waterline, then the use of either the Nye or LEFM models in prognostic simulations is unlikely to reliably simulate glacier behavior: the predicted absence of crevasses in compressional zones could prevent modeled retreat, or lead to punctuated episodes of retreat and temporary stabilization that result from unrealistic modeled patterns in crevassing.

Some of the physical processes present within the models tested here are undoubtedly important for ice fracture, even if they are not predictive in the forms tested within this study. For example, ice is known to have tensile strength, and therefore there is likely some threshold strain rate or stress below which crevasses will not form (see van der Veen (1998)). Inclusion of a non-zero threshold strain rate for crevasse formation decreases crevasse occurrence, even in places where they are observed. Thus, in the form presented here, the inclusion of a non-zero threshold strain rate for crevasse formation does not improve
model performance. More realistic performance is found with the LEFM model, which, similar to the Nye model with $\dot{\varepsilon}_{crit}$ >0, assumes that crevassing occurs only where the longitudinal stresses exceed a critical threshold for crevasse initiation (i.e., stress concentration > fracture toughness). However, the LEFM model takes into account the full stress tensor (via the effective stress) rather than just the along-flow longitudinal stress, and there are fewer crevassed regions where the LEFM model fails to predict crevassing. The incorporation of deformation enhancement and hydrofracture into the Nye model results in
comparable spatial patterns in crevassing for the LEFM and Nye models. However, there is no clear physical explanation for

the contrasting along-flow patterns in inferred enhancement, which suggest some glaciers have more viscous ice towards the terminus and others have less viscous ice towards the terminus. There are few observations of partial-ice thickness hydrofracture in Greenland to which we can compare our inferred water depths, but the modeled spatial patterns are unrealistic – they can vary by tens of meters over hundreds of meters along flow. Furthermore, approximately 1/4 of our observations were acquired prior to the onset of widespread seasonal surface melting. Because crevasses are known to drain over the course of the melt season (Everett et al., 2015; Lampkin and VanderBerg, 2014), we expect no water impounded in crevasses during spring. Therefore, the optimal deformation enhancement and water depth tuning parameters found here have no physical basis and should not be used to improve model agreement with observations.

Based on the comparison of observed crevasse depths with local strain rates, strain, and modeled crevasse depths, we hypothesize that our inability to reproduce small-scale (i.e., sub-kilometer) variations in observed crevasse depths using the Nye formulation stems from both its assumption of reduced stress concentration at crevasse tips in dense fields of crevasses and its ignorance of crevasse advection. As ice is advected into a stress field that favors crevasse formation, the depth to which a newly-formed crevasse penetrates depends on the instantaneous stress state as well as the micro- and macro-scale damage that the parcel of ice has inherited throughout its history (Bassis and Jacobs, 2013). If a crevasse penetrates deeper than its surrounding crevasses, then it will reduce the stresses on its neighbors, which will penetrate more shallowly than assumed by the Nye formulation (van der Veen, 1998). Propagation is favored at the deepest crevasses as they advect through extensional flow regimes, as supported by the observed along-flow increase in maximum crevasse depths at over half of our glaciers. Focusing of stresses within individual, deep crevasses is also supported by the slightly more realistic patterns in crevasse depth produced by the LEFM model, which uses the observed crevasse depths themselves to account for large-scale variations in stress concentration at crevasse tips. When either model is informed by a priori knowledge of crevasse depths, many of the large-scale spatial patterns in crevasse depths can be reproduced, but the simplifying assumption that crevasse depth is a function of the local stress state still results in model failure in regions of longitudinal compression.

The existing, local stress-dependent models for crevasse formation fail to simulate the complex patterns in observed, minimum crevasse depths at the tested glaciers. It is possible that true maximum fracture depths are uncorrelated with our open crevasse depth estimates. Such an occurrence would allow the modeled and true fracture depths to correlate in a manner that is at odds with our findings (e.g., Fig. 4). However, the modeled-observed crevasse depth disagreements highlighted here, including the observation of deep crevasses in regions with compressive strain rates, are problematic for a number of reasons. Unrealistic spatial variations in modeled crevasse depths may result in undue emphasis on the role of surface crevassing as a control on recent and future changes in terminus position. Our analysis of observed and modeled crevasse depths also suggests that advection of crevasses, and their associated mechanical and thermodynamic softening of ice, may exert an important control on the glacier stress balance. Confident projections of dynamic mass loss therefore require additional investigations on crevassing, including both remotely-sensed and in situ observations that track crevasse evolution through diverse stress (and

strain rate) regimes. We anticipate that these findings will spur novel efforts to model crevasse evolution, as well as the parameterization of calving in numerical ice flow models.

## Data Availability

The crevasse size distribution datasets constructed for this study are publicly archived at the Arctic Data Center (doi:
10.18739/A2WH2DF1F).

## Author Contributions

EME formulated the study, developed the code for the analysis, performed the majority of the data extraction, compilation, and analysis, wrote the initial draft of the manuscript, and led the revisions. TCB provided guidance on methodology, assisted
with data analysis, and revised the manuscript.

## Acknowledgements

This project is funded by NSF Office of Polar Programs collaborative awards 1933105 and 1714639 to E.M. Enderlin and 1716865 to T.C. Bartholomaus, as well as NASA grant NNX17AJ99G to Bartholomaus. We would like to thank Editor Stef
Lhermitte and the two anonymous reviewers for their recommended improvements to the analysis and its presentation in the manuscript.

## Competing Interests

The authors declare that they have no conflict of interest.

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

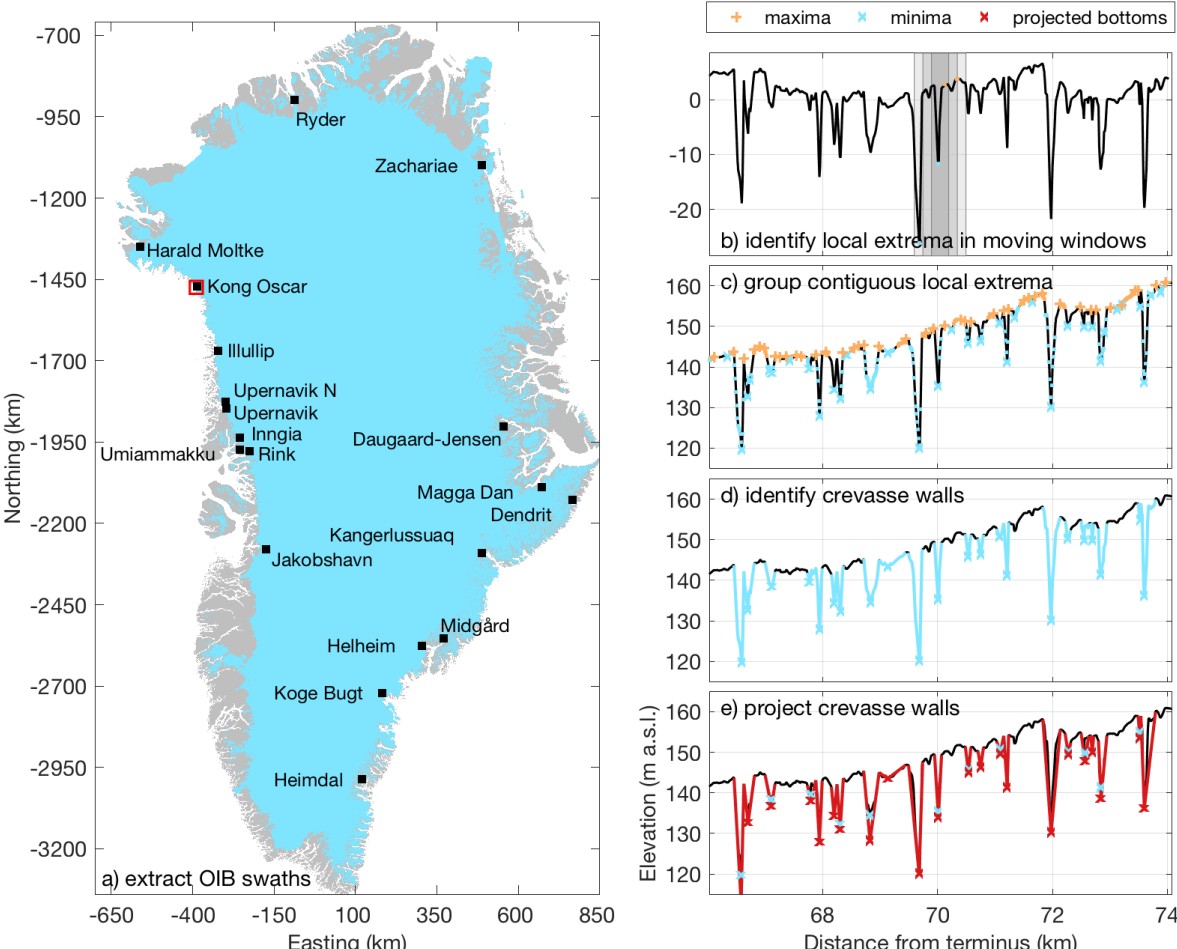

Figure 1: Map of glacier locations and example of the crevasse depth estimation approach applied to the elevation data for each glacier. a) Operation IceBridge transects (black squares) overlain on the Greenland Ice Mapping Project ice mask (light blue) and land mask (gray). Glacier names are from Bjork et al. (2015). The red box highlights the location of the profile in panels b-e. b) Moving window approach to find local extrema. The nested search windows (gray shading) and local extrema (colored points) overlain on the de-trended portion of the profile. Local extrema (blue dots) were filtered to c) isolate crevasse bottoms (blue x's) and top edges (orange +'s), d) locate steeply-sloped crevasse walls (blue lines), and e) project wall slopes to depth.

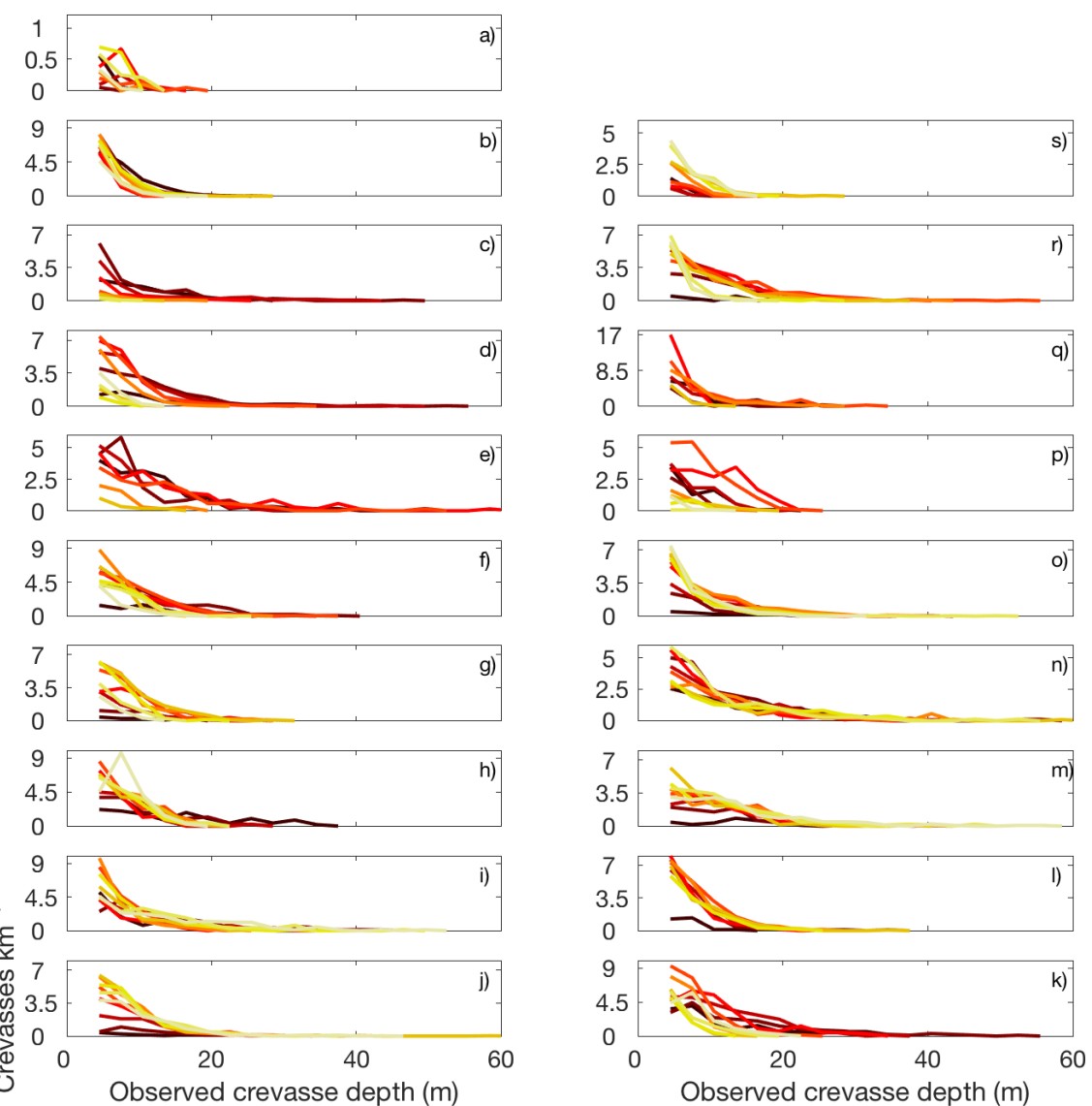

**Figure 2: Observed crevasse depth distributions for 1km-wide bins over the first 10km of each glacier. The distance from the terminus of each bin is distinguished by line color. Differences in area under the curves reflect variations in observed crevasse concentration between bins. Panels are geographically arranged so that western glaciers are on the left and eastern glaciers are on the right. Common names (Greenlandic names) are a) Ryder Gletsjer, b) Harald Moltke Bræ (Ullip Sermia), c) Kong Oscar Gletsjer (Nuussuup Sermia), d) Illiup Sermia, e) Upernavik North Isstrøm, f) Upernavik Isstrøm (Sermeq), g) Inngia Isbræ (Salliarutsip Sermia), h) Umiammakku Sermiat, i) Rink Isbræ (Kangilliup Sermia), j) Jakobshavn Isbræ (Sermeq Kujalleq), k) Heimdal Gletsjer, l) Koge Bugt Gletsjer, m) Helheim Gletsjer, n) Midgård Gletsjer, o) Kangerlussuaq Gletsjer, p) Dendrit Gletsjer, q) Magga Dan Gletsjer, r) Daugaard-Jensen Gletsjer, s) Zachariae Isstrøm.**

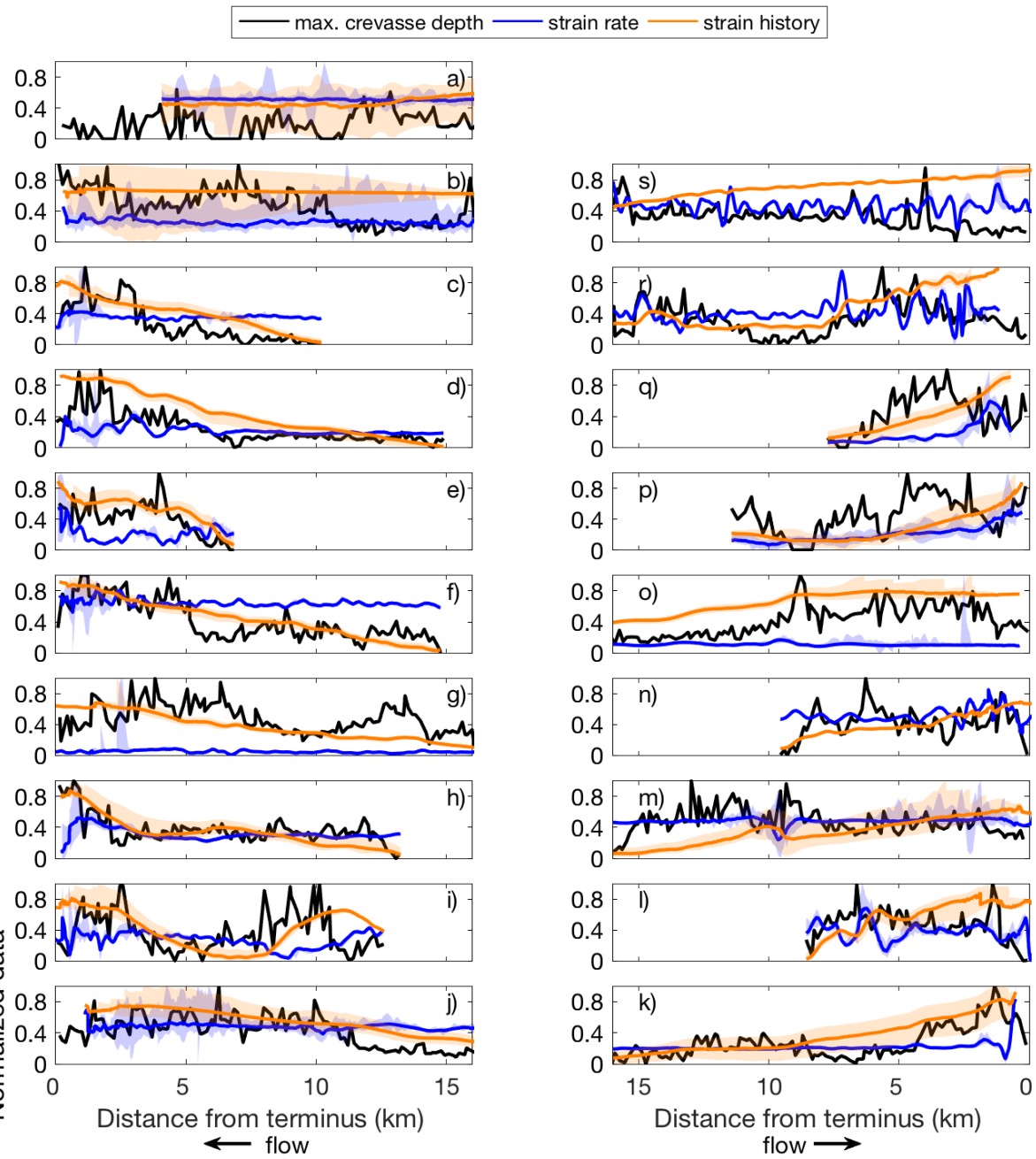

**Figure 3: Normalized profiles of maximum crevasse depth, local strain rate, and strain history. In each panel, the maximum crevasse depth in 150 m-wide bins is in black, the local strain rate is in blue, and the strain history is in orange. The median strain rate and strain history are shown as lines with shading indicating their temporal ranges. As in Fig. 2, the panels are geographically arranged.**

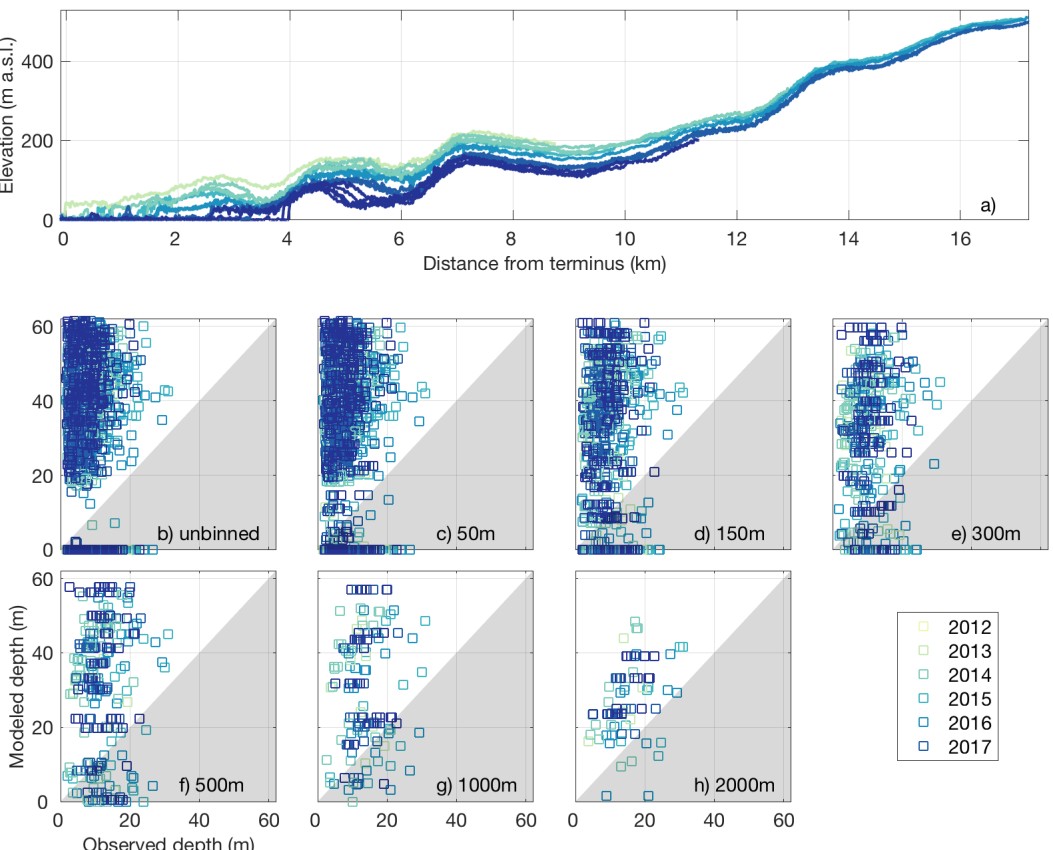

**Figure 4: Inngia Isbræ (Greenlandic Name: Salliarutsip Sermia) crevasse depth data. The legend indicates the observation year for all panels. a) Elevation profile time series extracted along the OIB swath. b-h) Scatterplots of observed crevasse depths plotted against crevasse depths modelled using the minimal Nye model. Points that fall in the white (gray) region represent model over-estimates (under-estimates) of observed depths. All observations are shown in b whereas the maximum observed and median modeled depths within along-flow bins are shown in c-h. The bin sizes in c-h (50-2000 m) reflect the range of spatial resolutions for numerical ice flow models.**


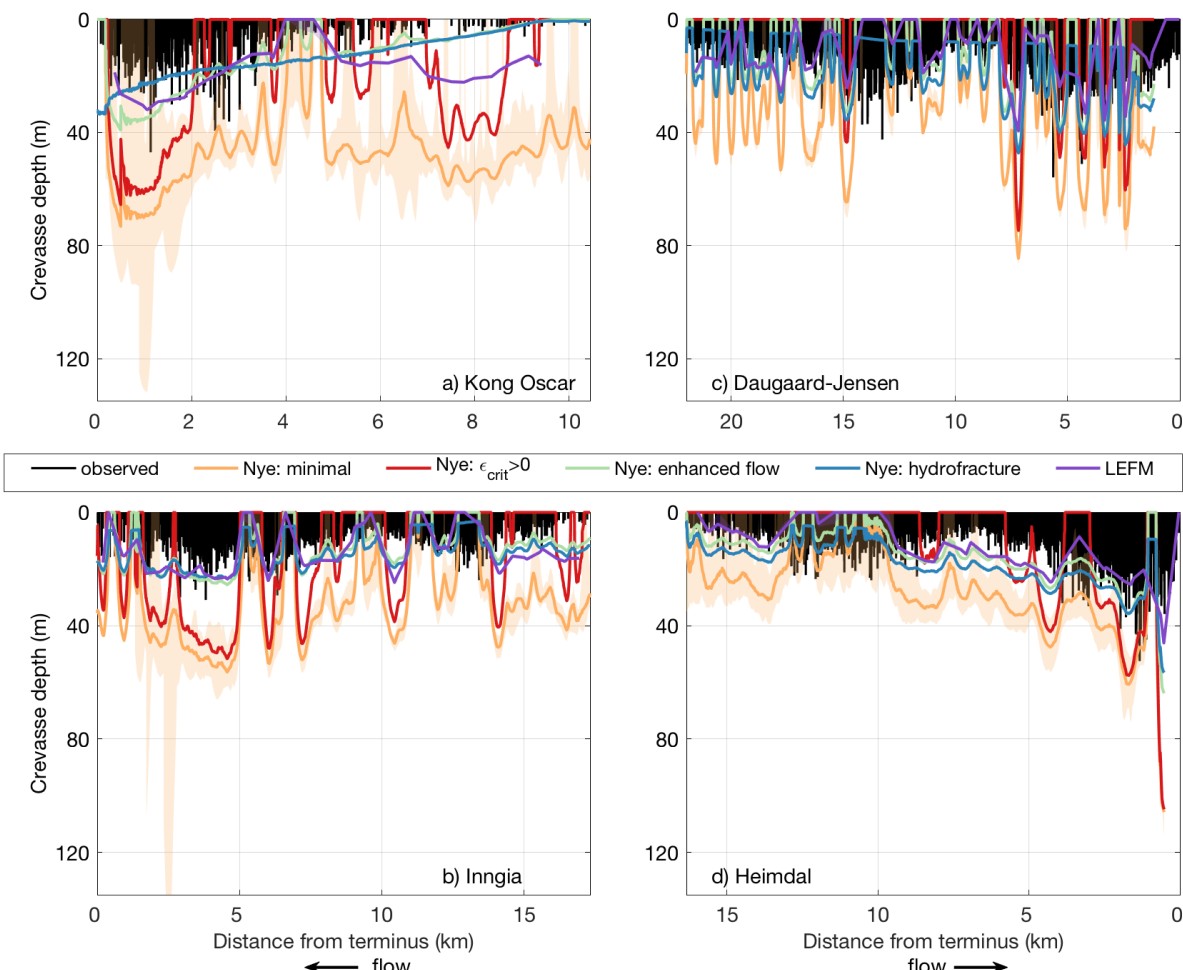


**Figure 5: Profiles of all observed crevasse depths (black lines) and modeled crevasse depths (colored lines) computed from the median velocity profile for a) Kong Oscar Gletsjer, b) Inngia Isbræ, c) Daugaard-Jensen Gletsjer, and d) Heimdal Gletsjer. Orange colors show the median (line) and temporal range (shading) in modeled crevasse depths using the minimal Nye formation (i.e., no critical strain rate, uniform viscosity, no water). The red, green, and blue lines show the Nye-modeled crevasse depths with**
**observation-based critical strain rates, flow enhancement, and flow enhancement with impounded water, respectively. The purple lines show the LEFM-modeled crevasse depths with the geometry-dependent stress intensity scaling factor calculated from observations.**

| Glacier Name | Latitude (°N) | Longitude (°E) | Max. Observed Depth (m) | Median Observed Depth (m) | Concentration (crevasses/km) | Max. Nye Depth (m) | Median Nye Depth (m) | Deformation Enhancement (unitless) | Max. Water Depth (m) | Max. LEFM Depth (m) | Median LEFM Depth (m) |
|---|---|---|---|---|---|---|---|---|---|---|---|
| Ryder | 81.7802 | -50.4556 | 10.9 | 4.8 | 1.0 | 29.4 | 15.2 | 0.64 | 6.1 | 5.1 | 5.1 |
| Harald Moltke | 76.5718 | -67.5659 | 21.1 | 3.3 | 15.2 | 34.8 | 22.2 | 0.63 | 10.2 | 11.9 | 9.7 |
| Kong Oscar | 76.0267 | -59.7052 | 47.0 | 5.0 | 9.6 | 69.7 | 46.6 | 0.79 | *32.7* | 28.7 | 23.1 |
| Illullip | 74.4026 | -55.9341 | 46.6 | 6.3 | 16.9 | 90.8 | 62.9 | 0.77 | *25.0* | 43.4 | 31.9 |
| Upernavik North | 72.9511 | -54.1183 | 59.9 | 8.6 | 17.8 | 118.5 | 52.2 | 0.70 | *30.7* | 56.3 | 40.4 |
| Upernavik | 72.8461 | -54.1578 | 36.3 | 7.6 | 19.6 | 69.6 | 41.4 | 0.58 | *10.8* | 33.8 | 20.8 |
| Inngia | 72.1022 | -52.5047 | 29.3 | 6.2 | 17.2 | 56.4 | 33.8 | 0.64 | 6.4 | 24.1 | 19.8 |
| Umiammakku | 71.7685 | -52.3880 | 35.3 | 6.9 | 15.2 | 64.7 | 39.1 | 0.55 | 9.4 | 32.0 | 28.9 |
| Rink | 71.7381 | -51.6096 | 31.6 | 5.9 | 21.9 | 72.3 | 49.7 | 0.62 | 13.1 | 34.3 | 23.5 |
| Jakobshavn | 69.1166 | -49.4560 | 58.6 | 7.3 | 17.9 | 72.3 | 61.9 | 0.67 | 28.1 | 34.7 | 24.6 |
| Heimdal | 62.8969 | -42.6730 | 24.0 | 5.4 | 18.5 | 33.8 | 28.2 | 0.58 | 12.7 | 18.8 | 14.9 |
| Koge Bugt | 65.2097 | -41.2156 | 35.1 | 5.9 | 17.3 | 106.3 | 58.2 | 0.76 | 8.7 | 41.0 | 27.1 |
| Helheim | 66.3941 | -38.3800 | 64.9 | 10.2 | 15.0 | 51.5 | 37.1 | 0.38 | 5.8 | 11.9 | 8.0 |
| Midgård | 66.5119 | -36.7300 | 55.4 | 9.3 | 19.0 | 108.2 | 56.8 | 0.54 | 14.0 | 33.2 | 21.8 |
| Kangerlussuaq | 68.5864 | -32.8397 | 50.0 | 4.5 | 17.8 | 80.8 | 45.9 | 0.70 | *27.9* | 40.4 | 18.6 |
| Dendrit | 69.3449 | -25.1687 | 23.9 | 6.5 | 11.4 | 52.8 | 35.2 | 0.62 | *7.3* | 21.3 | 13.2 |
| Magga Dan | 69.9375 | -27.1410 | 33.1 | 5.3 | 18.6 | 76.2 | 49.4 | 0.72 | 4.0 | 53.4 | 25.6 |
| Daugaard-Jensen | 71.8797 | -28.6788 | 55.9 | 7.2 | 16.2 | 84.5 | 38.1 | 0.53 | 12.0 | 35.8 | 24.2 |
| Zachariæ | 78.9161 | -21.0828 | 19.1 | 5.3 | 6.8 | 84.5 | 47.3 | 0.77 | 3.2 | 34.6 | 25.3 |

**Table 1: Observed and modeled crevasse characteristics within 5 km of the terminus. The name, location, maximum and median observed crevasse depths, median concentration of crevasses, maximum and median Nye-modeled crevasse depths, median tuned deformation enhancement factor, maximum tuned water depth, and maximum and median LEFM-modeled crevasse depths for each study site.**