# Peer review of "Sharp contrasts in observed and modeled crevasse patterns at Greenland's marine terminating glaciers"

_The Cryosphere, 2019_

## Referee Comment (RC1) · Anonymous Referee #1 · 19 Jul 2019

**Summary of manuscript**

This study infers crevasse depth from Operation IceBridge ATM data along swaths of 19 Greenland outlet glaciers over 6 spring campaigns, and compares these depths to the Nye formulation, a simple model for crevasse depth based on local stresses. The paper finds a systematic misfit between the "observed" and "modeled" crevasse depths: observations show consistently shallower crevasses than modeled. The authors speculate that the mismatch may originate from deformational history of the ice as well as non-uniform stress concentrations at crevasse tips. The authors conclude that the Nye formulation is inadequate for use in calving parameterizations.

[Figure]

**Comments**

The "observations" of crevasse depth presented here are flawed. The authors find a local elevation maximum, minimum, and maximum for each crevasse and calculate crevasse depth as the elevation difference – that is, they interpret the elevation minimum as the crevasse bottom. If the lidar were downward-looking, this could be a reasonable approximation; however, the ATM looks outward at $\theta = 15°$ angle, never straight down, in order to increase its effective swath width (see https://nsidc.org/sites/nsidc.org/files/technical-references/OIB-ATM-transceivers.pdf, where Table 1 notes the full scan angle, which is $2\theta$, for off-nadir angle $\theta$). This means that the ATM cannot see deeper than $\frac{W}{\tan\theta}$ ; for a crevasse $W = 10$ to $20$ meters wide, the ATM depth limitation is 40–70 meters, which is consistent with the results shown in Table 1. Crevasses may well be deeper than what the ATM instrument can constrain.

Although the above limitation is not mentioned in the manuscript, the authors do seem to account for the possibility that the ATM misses the crevasse bottom by introducing their V-shape correction. It is reasonable to expect that even a downward-pointing lidar might miss a crevasse bottom, since the beam footprint is ~1 meter, and crevasse width is much narrower than this at the tip, so the beam would see sidewalls as well as bottom and give a too-shallow average elevation. The authors thus infer a truer crevasse bottom by linearly extending the slopes of each wall to the point where they meet. While I appreciate this approach, it is still an underestimate of the crevasse depth. Although the large crevasses typically found on the outlet glaciers being studied here do appear, from the surface, to have wide V shapes, it is unlikely that the crevasses terminate at the tip of the apparent V. Stress concentration at that point are high, of course, which will drive the crevasse downward beyond the V tip. Crevasse shape below the V tip will be a narrow fissure, probably meandering irregularly downwards, as often seen on smaller crevasses upglacier or in alpine environments. This is

not captured by the authors' V-shape model, yet that "extra" depth is accounted for in the Nye formulation (as I understand it). Importantly, this fissure below the V tip would easily facilitate calving, which is the broad and important application of this study.

Overall, the full meaning and limitations of the observational data, described above, are not presented in the manuscript. The authors misinterpret ATM observations as crevasse bottoms, or as data that can be used to infer crevasse depths, when in fact the ATM measurements can only underestimate crevasse depths. True crevasse depths will be deeper, and may in fact compare well with the Nye formulation. The data presented here cannot support evaluation of the Nye formulation. Thus, much of the manuscript, including its title, is substantially flawed. Most of the tuning analysis, for instance, is irrelevant, given the above.

The figures were somewhat difficult to interpret. An illustration of data from a single crevasse, or from a few crevasses across a reach of a kilometer or two, would have greatly helped me understand the picking, extrapolation, and manual/automated approach. I liked the organization of the panels for each glacier within a rough map/array of Greenland (Figures 2–3), but I missed helpful features like titles, y-axis labels, and consistent y-axis ranges among panels.

---

## Author Comment (AC1) · 20 Jul 2019

We thank referee #1 for his or her timely and engaging review. We respond here to a couple of points that might have been under-appreciated in the referee's review of our manuscript, prior to a more complete revision and response that we plan to complete following receipt of the 2nd referee's comments. In the revision, and guided by the advice and critique of referee #1, we will be certain to strengthen the discussion of the following elements of our analysis.

First, the referee points out that the off-nadir pointing angle of the OIB laser is a potential limitation of laser-measured crevasse depths. While the referee is correct in her

or his description of this limitation, off-nadir pointing limits measurements of crevasse depth, but does not bias our measurements of crevasse shape. If the off-nadir path of the laser beam intersects a crevasse wall or crevasse lip, it will simply not return elevation measurements of the crevasse bottom. In this sense, the off-nadir angle joins two other limitations, broken serac debris and ponded water at the bottom of crevasses, as a limiting factor in assessing crevasse depths. Thus, our analysis of crevasse morphology and the finding that crevasses are dominantly "V" shaped addresses this broader concern, since, guided by our data, we extrapolate crevasse walls to the bottom tip of the V.

Second, although the crevasses are very well-fit with V shapes, we concur with the referee that the true, maximum depth of many crevasses undoubtedly extends beneath the tip of the V as a paper-thin, englacial fracture. Thus, our measurements are minimum crevasse depths. This is a point we will make more explicitly in our revision. However, our result remains: even if we apply a scalar (either additive or multiplicative) factor to the observed crevasse depths to minimize the average bias with respect to the depths predicted using the Nye formulation, the patterns in the depths extrapolated from elevation observations and the patterns inferred from velocity fields using the Nye formulation do not agree. We find no correlation between observed and predicted crevasse depths (Fig. 4), and the spatial pattern of observed crevasse depths is unrelated to the spatial pattern of predicted crevasse depths (Fig. 5). We cannot envision some other, plausible correction factor that will bring Nye-modeled crevasse depths in line with observed crevasse depths. As ours is the most complete and thorough examination of the popular Nye model, building on the foundational work of Mottram and Benn, 2009, we believe our results are highly significant. We repeat for emphasis that our results are deeper than the finding of a systematic offset between modeled and observed crevasses. Instead, we find no relationship whatsoever.

The discrepancy in crevasse depth patterns is important because it tells us that we cannot rely on local strain rates to accurately model crevasse depths. If we use surface

crevasse depths from the Nye model in numerical models of calving, as has been done in a number of studies, then the model may fail to capture the correct spatio-temporal evolution of terminus position. If Nye-modeled crevasses are used to route surface water to the bed, the spatio-temporal evolution of these water inputs are likely to be incorrect. We will revise the manuscript to address these points following the receipt of all reviews, emphasizing that although our observed crevasse depths may under-estimate true depths, the discrepancies in the observed and modeled depths suggest that the predictive power of the Nye formulation for crevasse depths is limited when applied to the complex histories and geometries of Greenland's outlet glaciers.

We eagerly anticipate the comments of the 2nd referee, and the suggestions of the editor.

---

## Referee Comment (RC2) · Anonymous Referee #2 · 30 Sep 2019

In this paper the authors compile a new dataset of crevasse depths derived from lidar swaths made by IceBridge flights over 19 different glaciers in Greenland. They compare their very extensive observations with a commonly used crevasse depth model introduced by Nye and based on strain rates derived from satellite velocities. They conclude that the Nye model that has been used as a calving criterion is flawed as it appears to overestimate crevasse depths compared to observations in this study. This is a really interesting and well-written paper and I thoroughly enjoyed reading it. In many ways, the finding that the Nye model does not work very well is almost the least interesting outcome of this research, and in fact this is also one of the less novel findings, nonetheless the persistent use of the Nye model as a fracture criterion suggests that the ice modelling community has yet to absorb this message! I very much appreciated the results and discussion sections which raised many familiar points related to the difficulties in translating velocities to strain rates and stresses as well as attempting to refine the Nye Depth model. It's likely beyond the scope of this work but I look forward to future work examining the implications for a fracture mechanics approach to crevasses based on or related to the present study. The authors have done a huge amount of work and their crevasse depth dataset as well as the lidar techniques they have developed will surely be of great value for future research into the problems and processes of ice fracture on outlet glaciers. Given the hazards of measuring and observing crevasses in the field, the development of these kind of remote sensing techniques are crucial for advancing the field and many of their findings are replicating similar findings from field studies albeit on a smaller scale. Some further specific comments: 1) Perhaps the main issue with the study is the assumption that the lidar technique actually reaches the bottom of the crevasses. As extensive discussion in Colgan et al (2016) and van der Veen (1999) ( a missing reference that would perhaps help to interpret their results) also showed, defining the "bottom" of a crevasse is surprisingly difficult. The authors assume a v-shape is the standard shape, but field studies indicate that the fracture often penetrates substantially further than is easy to measure with plumb lines or laser scanning techniques and the reported measured depths are usually best understood therefore as a minimum depth. A better model of crevasse shape may be an initial v-shape (partly also due to ablation processes) that narrows but extends with sides almost parallel to each other for some distance. Observations of fractures from within ice caves would also support this shape. There is no reason to believe that the lidar technique would not suffer from similar issues as manual measurements and while I think it would be unreasonable to redo all the calculations with a different profile since we don't have a clear idea what this might be, I think it would be helpful to discuss the implications if the v-shape model turns out to be an oversimplification. The authors to their credit do discuss this to some extent but their explanation on lines 110 to 124 is a little unclear and should be expanded further,

also when considering the results. 2) Similarly the authors mention falsely identified crevassses (line 105) but it is not clear how this was done. Please expand on this and roughly how many were positive/negative. 3) In the crevasse depth comparison with modelled depths section (3.2) there are several statements that are a little confusing; on line 244, it is stated that the model failed to predict crevasses in compressional zones, but Nye's model explicitly only calculates crevasse depths where there is a positive (extensional strain) so this is not surprising – was there an extensional strain across glacier rather than down glacier in these zones that could be used? Other studies have also used the strain ellipse rather than simply the longitudinal strain rate. 4) The acceleration of an outlet glacier does not necessarily lead to enhanced longitudinal strain rates across the whole glacier, did the acceleration of Inngia Isbræ lead to a significant increase in strain rates (Line 254)?

Minor comments: Line 17: Surely this should be "are both affected by and affect" Line 60: Given that the crevasses were mostly around 6m in depth I don't think a water depth of 1 to 10 m can be described as "small" – sorry a picky point... Line 90-91: Sentences that use brackets to denote the reverse condition are really hard to read, it's better to write this as two separate sentences (alsu required by e.g. AGU style guide). Line 155: remove "a"

---

## Author Comment (AC2) · 17 Oct 2019

Response to Reviewer 2: We thank the 2nd reviewer for their careful review and for their helpful suggestions in improving the clarity and impact of our manuscript. We appreciate that the reviewer finds our work interesting, novel, and compelling, and is convinced by our main conclusions: that local strain rates are unrelated to the pattern of crevasses found along 19 Greenland outlet glaciers. This second review also notes some of the foundational work we cite that came to similar conclusions, albeit with a more modest dataset (that by Mottram and Benn, 2009), and we thank the reviewer for drawing our attention back to the work of Colgan et al., 2016, and van der Veen, 2009.

[Figure]

We intend to reference their discussion of crevasse depths and their V-shapes in our revision as well as Liu et al., 2015 (doi:10.1007/s11430-013-4796-x), who estimated crevasse depths in Antarctica using ICESat data and the assumption of V-shaped geometries.

As with the first-submitted review, the second reviewer's dominant concern (specific comment 1) is that the V-shaped bottom estimates are likely to be minimum bounds. In reality, fractures likely extend below the bottom of V-shaped crevasses, although they are nearly impossible to measure over any kind of significant spatial scale. This is an important point that we plan to make more forcefully and directly in our revision. As pointed out by the reviewer, we already discuss the implications of the V-shaped assumption to some extent on lines 110-124, but we will expand on this discussion to make the potential under-estimation of fracture depths more explicit. However, it is important to note that even if fractures penetrate below the 'open' portion of crevasses, this caveat does little to alter the main thrust of our work–that the Nye model fails to reproduce observed patterns of crevasse depths. For example, in our Figure 4b-h, we show the mis-match between our observed crevasse depths and modeled crevasse-depths. One of the most important messages of this work is that even if we augment the crevasse depth model so that the average modeled depths are similar to observations, the spatial variations in observed depths still cannot be explained by the model. Modeled depths are not predictive of observed depths and no plausible modification of the observed crevasse depth data will bring the observations into accord with the model. We will modify the manuscript so that the potential under-estimation of crevasse depths due to the assumption of V-shaped geometries is more clear and to emphasize that this potential under-estimation does not alter our primary finding: crevasse depth models that rely only on local strain rates cannot reproduce observed crevasse depths.

In addition, the second reviewer asks for clarification at several points, and suggests a number of grammatical modifications. We expect that our resolution of these points will increase comprehension and impact. Specifically, with regard to the points of clarification, regarding specific comment 2, these falsely identified crevasses are crevasses expected to occur in the model, but that are absent from the observations. Regarding comment 3, at this point, we only had intended to reinforce the fact that the Nye model does not predict crevasses in compressional zones, as the reviewer writes. Regarding comment 4, Ingia Isbrae did undergo widespread increase in strain rates associated with its terminus retreat. We propose to clarify and expand on each of these points in our revision.

In conclusion, we appreciate the care and interest expressed by the second reviewer. They have raised important points that do not challenge the central conclusions of our study, but will improve clarity of the manuscript when addressed in revision.

Response to the Editor and both reviewers: Dear Editor, We have now received two constructive reviews of our manuscript. Both reviewers raised important methodological questions about our ability to measure the bottom of a crevasse by measuring or extrapolating the bottom of its V-shaped surface expression. We feel that this concern is an important one, and we look forward to addressing it through the addition of modest qualifying text. However this concern does not challenge our central conclusion: crevasse depth models that rely only on local strain rates cannot reproduce observed crevasse depths, even when tuned to minimize the bias in crevasse depths. As we and the reviewers have pointed out, a more limited study of a glacier in Iceland has come to a similar conclusion (Mottram and Benn, 2009), however, to quote reviewer 2, " the persistent use of the Nye model as a fracture criterion suggests that the ice modelling community has yet to absorb this message!" Local stresses, associated with local strain rates, are not correlated with locally observed crevasses. The Nye model is insufficient to reproduce crevassing, and therefore it is insufficient to model iceberg calving either.

In a revised submission, we are confident that we can address this important concern without dulling the impact of our paper. We also look forward to incorporating suggested changes to the text, clarifying the places where our intent had been unclear,

and reformatting several of the figures.

Thank you for your consideration and we look forward to receiving your direction regarding the next steps of review.

Sincerely, Ellyn Enderlin and Timothy Bartholomaus
* * *

---

## Author Response (AR1)

Dear Editor,

We would like to thank you and the two anonymous reviewers for your comments on our manuscript. The primary concern of both reviewers was our under-estimation of observed crevasse depths due to our assumption that crevasse geometries are approximately V-shaped. In the attached revision, we have more explicitly acknowledged the limitations of this assumption and have emphasized that, despite the potential under-estimation of observed crevasse depths, the major results of our analysis hold true: current crevasse depth models that rely solely on local stresses cannot reliably simulate observed patterns in crevasse depth. While relaxation of the V-shaped crevasse assumption potentially deepens our observed crevasses towards the depths modeled using the Nye 'zero-stress' crevasse depth model, and tuning of the models can shallow the modeled depths towards the observed depths, the pattern of predicted crevasse depths is fundamentally at odds with the observed pattern of crevasse depths. We make this point more directly in our revision.

To further emphasize this point and provide a more compelling study, we have added a new analysis to complement the existing analysis of the Nye model for crevasse depths. In this new analysis, we apply the linear elastic fracture mechanics (LEFM) model to our study glaciers, and compare these modeled crevasse depths with our observed crevasse depths as well. The LEFM model out-performs the Nye model when informed by the crevasse depth observations, but both models have inconsistent accuracy across the broad range of conditions observed for Greenland's fast-flowing glaciers. In response to your suggestions and those of our reviewers, we feel that the manuscript is greatly improved and look forward to receiving your decision.

Sincerely,
Ellyn Enderlin & Tim Bartholomaus

Reviewer comments in black, responses in blue.

**Reviewer 1**
**Summary of manuscript**
This study infers crevasse depth from Operation IceBridge ATM data along swaths of 19 Greenland outlet glaciers over 6 spring campaigns, and compares these depths to the Nye formulation, a simple model for crevasse depth based on local stresses. The paper finds a systematic misfit between the "observed" and "modeled" crevasse depths: observations show consistently shallower crevasses than modeled. The authors speculate that the mismatch may originate from deformational history of the ice as well as non-uniform stress concentrations at crevasse tips. The authors conclude that the Nye formulation is inadequate for use in calving parameterizations.

**Comments**
The "observations" of crevasse depth presented here are flawed. The authors find a local elevation maximum, minimum, and maximum for each crevasse and calculate crevasse depth as the elevation difference – that is, they interpret the elevation minimum as the crevasse bottom. If the lidar were downward looking, this could be a reasonable approximation; however, the ATM looks outward at $\theta = 15°$ angle, never straight down, in order to increase its effective swath width (see https://nsidc.org/sites/nsidc.org/files/technical-references/OIB-ATMtransceivers.pdf, where

Table 1 notes the full scan angle, which is 2θ, for off-nadir angle θ). This means that the ATM cannot see deeper than W tan θ ; for a crevasse W = 10 to 20 meters wide, the ATM depth limitation is 40–70 meters, which is consistent with the results shown in Table 1. Crevasses may well be deeper than what the ATM instrument can constrain.

Although the above limitation is not mentioned in the manuscript, the authors do seem to account for the possibility that the ATM misses the crevasse bottom by introducing their V-shape correction. It is reasonable to expect that even a downward-pointing lidar might miss a crevasse bottom, since the beam footprint is ~1 meter, and crevasse width is much narrower than this at the tip, so the beam would see sidewalls as well as bottom and give a too-shallow average elevation. The authors thus infer a truer crevasse bottom by linearly extending the slopes of each wall to the point where they meet. While I appreciate this approach, it is still an underestimate of the crevasse depth. Although the large crevasses typically found on the outlet glaciers being studied here do appear, from the surface, to have wide V shapes, it is unlikely that the crevasses terminate at the tip of the apparent V. Stress concentration at that point are high, of course, which will drive the crevasse downward beyond the V tip. Crevasse shape below the V tip will be a narrow fissure, probably meandering irregularly downwards, as often seen on smaller crevasses upglacier or in alpine environments. This is not captured by the authors' V-shape model, yet that "extra" depth is accounted for in the Nye formulation (as I understand it). Importantly, this fissure below the V tip would easily facilitate calving, which is the broad and important application of this study.

Overall, the full meaning and limitations of the observational data, described above, are not presented in the manuscript. The authors misinterpret ATM observations as crevasse bottoms, or as data that can be used to infer crevasse depths, when in fact the ATM measurements can only underestimate crevasse depths. True crevasse depths will be deeper, and may in fact compare well with the Nye formulation. The data presented here cannot support evaluation of the Nye formulation. Thus, much of the manuscript, including its title, is substantially flawed. Most of the tuning analysis, for instance, is irrelevant, given the above.

The figures were somewhat difficult to interpret. An illustration of data from a single crevasse, or from a few crevasses across a reach of a kilometer or two, would have greatly helped me understand the picking, extrapolation, and manual/automated approach. I liked the organization of the panels for each glacier within a rough map/array of Greenland (Figures 2–3), but I missed helpful features like titles, y-axis labels, and consistent y-axis ranges among panels.

The reviewer is correct that our method is only capable of mapping the open portion of crevasses and we have made that more explicit in several locations throughout the text (lines 169, 235-239, 297-320, 328-329, and in the abstract and discussion). We also have added text starting at line 297 that explicitly states that the off-nadir viewing geometry may prevent the extraction of crevasse bottom elevations from narrow crevasses. However, we disagree with the reviewer that these limitations prevent the estimation of surface crevasse depth from the observations. As pointed out by reviewer 2, the most interesting part of this study is not that the Nye model apparently over-estimates crevasse depths but that it cannot reproduce patterns. Even though we likely under-estimate the true depth of fractures beneath crevasses, if the models are useful, the along-flow patterns in crevasse depths should be similar in our minimum depth estimates and the

true depths. In addition to our extensive revisions to acknowledge the methodological limitations of our study, we have reframed the text to focus more on the disagreement in crevasse depth patterns. We have also added modeled depths using the linear elastic fracture mechanics (LEFM) formulation for crevasse depths from van der Veen (1998) to emphasize that the disagreement between modeled and observed depths stems from the assumption that crevasse depths are governed by only the local stress state The title of the manuscript has been changed to account for these revisions.

We refer the reviewer to Figure 1 for observations of crevasse depths over a relatively small subregion. The subplots for the observed and modeled crevasse depths in Figures 5 and S23 now have standardized y-axes, as recommended by the reviewer. We did not, however, standardize the y-axes in Figure 2 since this would obscure the along-flow patterns in crevasse concentration for the glaciers with relatively sparse crevassing. We have not added titles to the figures, in keeping with the formatting conventions of *The Cryosphere*.

**Reviewer 2**

In this paper the authors compile a new dataset of crevasse depths derived from lidar swaths made by IceBridge flights over 19 different glaciers in Greenland. They compare their very extensive observations with a commonly used crevasse depth model introduced by Nye and based on strain rates derived from satellite velocities. They conclude that the Nye model that has been used as a calving criterion is flawed as it appears to overestimate crevasse depths compared to observations in this study. This is a really interesting and well-written paper and I thoroughly enjoyed reading it. In many ways, the finding that the Nye model does not work very well is almost the least interesting outcome of this research, and in fact this is also one of the less novel findings, nonetheless the persistent use of the Nye model as a fracture criterion suggests that the ice modelling community has yet to absorb this message! I very much appreciated the results and discussion sections which raised many familiar points related to the difficulties in translating velocities to strain rates and stresses as well as attempting to refine the Nye Depth model. It's likely beyond the scope of this work but I look forward to future work examining the implications for a fracture mechanics approach to crevasses based on or related to the present study. The authors have done a huge amount of work and their crevasse depth dataset as well as the lidar techniques they have developed will surely be of great value for future research into the problems and processes of ice fracture on outlet glaciers. Given the hazards of measuring and observing crevasses in the field, the development of these kind of remote sensing techniques are crucial for advancing the field and many of their findings are replicating similar findings from field studies albeit on a smaller scale.

We thank the reviewer for her or his message, and appreciate that she or he has grasped the important implications of our study. We have sought, in the revision, to bring our these conclusions even further.

**Some further specific comments:**
1) Perhaps the main issue with the study is the assumption that the lidar technique actually reaches the bottom of the crevasses. As extensive discussion in Colgan et al (2016) and van der Veen (1999) ( a missing reference that would perhaps help to interpret their results) also showed, defining the "bottom" of a crevasse is surprisingly difficult. The authors assume a v-shape is the

standard shape, but field studies indicate that the fracture often penetrates substantially further than is easy to measure with plumb lines or laser scanning techniques and the reported measured depths are usually best understood therefore as a minimum depth. A better model of crevasse shape may be an initial v-shape (partly also due to ablation processes) that narrows but extends with sides almost parallel to each other for some distance. Observations of fractures from within ice caves would also support this shape. There is no reason to believe that the lidar technique would not suffer from similar issues as manual measurements and while I think it would be unreasonable to redo all the calculations with a different profile since we don't have a clear idea what this might be, I think it would be helpful to discuss the implications if the v-shape model turns out to be an oversimplification. The authors to their credit do discuss this to some extent but their explanation on lines 110 to 124 is a little unclear and should be expanded further, also when considering the results.

We have added text in a number of locations (see the response to reviewer 1 above) to be more explicit in the limitations of the methodology used here. We agree with the reviewer that it does not make sense to introduce a more complex shape since that shape would be unconstrained by observations, and the 'V' shapes are consistent with observations. Instead we emphasize that our crevasse depth estimates should be treated as minimum estimates since they cannot account for micro fractures that may extend meters, or even tens of meters, below the open portion of each crevasse. In addition to more explicit acknowledgement of the potential under-estimation of observed crevasse depths, we have slightly reframed the manuscript to focus more on the disagreement in observed and modeled crevasse depth patterns rather than focusing on the magnitude differences.

2) Similarly the authors mention falsely identified crevasses (line 105) but it is not clear how this was done. Please expand on this and roughly how many were positive/negative.

The false positives and negatives were identified based on a comparison with the manually-delineated dataset. We have added the following to the text: "For these optimal window sizes, the median false negative rate was 1.2% and the median false positive rate was 38.5% across all glaciers. In other words, the automated algorithm missed ~1% of manually-identified crevasses and identified ~38% more crevasses than the manual interpreter."  It is clear that the algorithm is somewhat more liberal in its definition of "crevasse" than the human analyst, however, in review, we have found these algorithmically identified, "extra" crevasses to also be consistent with our expectations for crevasse geometry.

3) In the crevasse depth comparison with modelled depths section (3.2) there are several statements that are a little confusing; on line 244, it is stated that the model failed to predict crevasses in compressional zones, but Nye's model explicitly only calculates crevasse depths where there is a positive (extensional strain) so this is not surprising – was there an extensional strain across glacier rather than down glacier in these zones that could be used? Other studies have also used the strain ellipse rather than simply the longitudinal strain rate.

We agree that this is not a surprising result given that the Nye and LEFM models assumed Mode 1 failure, but felt that it is worth emphasizing in this paper given the use of the crevasse penetration depth calving parameterizations in numerical models. The LEFM model inherently

accounts for the full stress tensor, as it is dependent in part on the effective stress, yet there are few places where the Nye model does not predict any crevassing but the LEFM model predicts that crevasses are present (see Figure S23). However, we note in the text that the consideration of the full stress tensor reduces modeled regions of no crevassing by the LEFM model relative to the Nye model. This inter-model comparison suggests that inclusion of cross-flow longitudinal extension in the Nye model may reduce the spatial extent of model-predicted crevasse-free zones but is unlikely to drastically improve the ability of the model to reproduce spatial patterns in crevasse depth.

4) The acceleration of an outlet glacier does not necessarily lead to enhanced longitudinal strain rates across the whole glacier, did the acceleration of Inngia Isbræ lead to a significant increase in strain rates (Line 254)?

As shown by the narrow bands of blue and orange shading for the temporal variations in the local strain rates and cumulative strain, respectively, Inngia Isbræ did not appear to undergo substantial temporal variations in strain rate over the observation period. We suspect that the dynamic thinning observed from 2012-2017 was initiated by a change in the glacier stress balance prior to 2012, which would not necessarily appear as a change in strain rate over the observation period.

**Minor comments:**
Line 17: Surely this should be "are both affected by and affect"
Changed

Line 60: Given that the crevasses were mostly around 6m in depth I don't think a water depth of 1 to 10 m can be described as "small" – sorry a picky point...
Removed

Line 90-91: Sentences that use brackets to denote the reverse condition are really hard to read, it's better to write this as two separate sentences (also required by e.g. AGU style guide).
Changed

Line 155: remove "a"
Changed

[revised manuscript text omitted]

---

## Referee Report (RR1)

**Review** of "Local stress models do not predict observed crevasse patterns at Greenland's marine terminating glaciers" by Enderlin and Bartholomaus, for *The Cryosphere Discussions*

**Summary of manuscript**

This manuscript describes a new, large dataset of lidar-derived estimates of crevasse depths on Greenland outlet glaciers and compares these depths to calculations by two simple, physically based models for crevasse depths. As in its earlier draft, the manuscript reports that the data-derived and model-based depths are "unrelated" and thus crevasse depth models have questionable utility. Compared to the earlier draft, this version briefly acknowledges the shortcomings in the central dataset, and adds comparison with a second crevasse depth model.

**Concerns about validation**

New datasets, particularly ones as large in scope as the one presented here, are typically validated before they are used. Unfortunately, no validation was done for this dataset. Therefore, it is hard to have confidence in the authors' conclusions that local stress models are inaccurate. This is my first major criticism with the work presented.

The authors' justification of their V-shape model (lines 131–138) remains problematic. They advocate that a crevasse initially forms as a V, and later processes, like serac toppling, cause deviations from a V. This manuscript assumes "negligible fracture extent beyond the bottom of the V", but as I pointed out in my previous review, crevasses in lower surface stress regimes are more irregular than a V and tend to quickly go off-nadir with depth. For example, Figure 1a in Smith et al. (2015) is a field photo that shows a near-surface cross section of a Greenland crevasse in a lower-stress setting. The surface expression (a V, rough depth estimate 3 meters based on the scale of the person in the background) gives way to a curved, inclined fracture of at least the same depth beneath the tip of that V. For the outlet glacier crevasses the authors here are studying, the deviatoric stresses are higher and the V shape extends much deeper, but below the tip of the V, where the stresses decay, this field photo should be representative. Clearly, beneath the tip of the V lies a narrow, fractured plane that is not measureable by a lidar with a 1 meter footprint.

The authors attempt to quantify the ability of the lidar to locate the "true crevasse bottom" by comparing up-glacier to down-glacier swaths (lines 155–158). Unfortunately, this does not actually evaluate the accuracy of the lidar, merely its repeatability. The 1 meter footprint of the lidar is too large to measure sub-centimeter fractures, and this will be the case regardless of the direction of flight or the time of day.

The revision contains a few added sentences (lines 85-89) that address (but does not put to rest) the above concern.

"While the full crevasse depth is unmeasurable from glacier surfaces, we assume here that the depth of the visible, near-surface void space is positively correlated with the full depth of the crevasse, which likely extends beyond the depth of the void space. We refer to the maximum, visibly open depth below the surface as the observed crevasse depth."

First, "observed crevasse depth" is misleading, since full crevasse depth is not actually being observed. I suggest "observable depth" or "observed minimum depth", which I perceive as more accurate.

More importantly, this assumption needs to be tested (the dataset validated). It may well be the case that the depth of the V is correlated to the full depth of the crevasse. However, it may not. It is unreasonable to discredit long-standing theories with an unvalidated dataset that the authors "assume" is correlated with truth.

**Concerns about correlation analysis**

The authors have reframed some of their discussion to emphasize the lack of correlation between "observed" crevasse depth and modeled or expected crevasse depth. However, a lack of correlation should be the expected result, due to the two significant limitations inherent to the dataset: (1) scan angle limitations described in my first review, and (2) the false approximation of a V-shape described above. For both these reasons, and as now mentioned in the manuscript (lines 131–138), the crevasse depth "observations" are minima.

An attempt to compare truncated depth "observations" with model-based predictions that span a full range of depths will clearly return no correlation. This is indeed what the authors find, yet they attribute this to deficiencies in the models.

The above forms my second major concern with the analysis: the known bias in the dataset renders it (as currently presented) unable to test the accuracy of any models. One idea the authors might pursue to address limitation (1) above is selecting only crevasses whose sensed depth-to-width ratios are within the range of the lidar.

**Review conclusions**

Overall, the dataset requires validation, and the manuscript needs to more responsibly represent the limitations of the "observed depth" dataset throughout the manuscript. It is evident that a lot of work went into producing this dataset. Therefore, I hope that the authors can think deeper into the limitations of the data, then validate the dataset so that it may be used for analysis.

**References**

Smith, L. C. et al. (2015), Efficient meltwater drainage through supraglacial streams and rivers on the southwest Greenland ice sheet, Proceedings of the National Academy of Sciences, 112(4), 1001–1006, doi:10.1073/pnas.1413024112.

---

## Referee Report (RR2)

**Review** of "Sharp contrasts in observed and modeled crevasse patterns at Greenland's marine terminating glaciers"
by Enderlin and Bartholomaus, for *The Cryosphere Discussions*

**Summary of manuscript**

This iteration of the Greenland airborne crevasse depth analysis acknowledges the limitations to the observations. The analysis is mostly the same as presented before, with additions to the discussion and changes to the conclusions. The new conclusion – that advection and/or blunting influence crevasse depth but are not included in the usual simple models tested here – is reasonable, but strangely does not appear in the abstract. The revision still does not incorporate the known uncertainties in the crevasse depth dataset, which therefore still looks like noise to me. I suggest that the authors remove too-narrow crevasses (where the ATM cannot sense the tip of the V) from their dataset – perhaps I should have been more specific, in previous reviews, that this should have been done. This should be simple to implement using the data the authors already have.

Overall, the paper is improved from its earlier versions, but suffers from new issues of clarity and organization, and lingering science problems.

**Major comments**

1. The description of the modeling analysis is convoluted and spread across multiple sections. Equations are presented in the Introduction and built on in the Methods. A formula for the LEFM model is never given; I think it is intertwined with the Nye formulation, but I am not sure. A standard approach would define modeled crevasse depth $d_X =$ for each model $X$ in the Methods. This would make it easy to compare the factors and adjustments that each model incorporates.

2. The abstract ends with a punt: "We therefore suggest that additional analyses of the controls on crevassing are performed prior to implementation of either crevassing model within ice sheet models." It seems that the authors land on advection (line 412) as the source of the misfit between estimated and modeled crevasse depth, although they do not quantitatively explore it. Adding the advection hypothesis to the abstract, and supporting it with a back of the envelope calculation in the paper itself, would strengthen it.

   - Advection estimate: Figure 5a, Nye Minimal or Nye Critical model, shows compressive regions up to ∼1 km long. With estimated flow speeds 1 km/yr, this means that crevasses that form upstream must persist for ∼1 year on their transit through compressive areas. This is an entirely reasonable lifetime for a crevasse, even on an outlet glacier. This could be supported with observations (e.g., tracking in satellite images) or cited (I am not sure of a reference) or left as a statement (which I would believe).

3. Deformation enhancement factor: The name is misleading. The term $D$ is presented as a reduction in viscosity due to fractures, roughly in line with Borstad et al. (2016). However, Borstad did a full L-curve analysis to balance smoothness in their $D$ field against the misfit

that $D$ attempts to minimize. What is presented here, on the other hand, is pure misfit (Equation 4). It is disingenuous to calculate misfit and name it something else, effectively ascribing all of the misfit to a particular process. The authors do seem to acknowledge this: "such detailed tuning is neither physically motivated nor practical" (line 230) and "the optimal deformation enhancement and water depth tuning parameters found here have no physical basis" (line 387), but they continue with their analysis anyway. Unfortunately, this sort of dismissing of warning signs is a recurring pattern in the analysis.

4. A priori information on crevasse depth: The authors fold their crevasse depth observations into the LEFM model (line 251), which they find improves the agreement with the data. Of course it does. However, they need not (and should not) have done this, as the LEFM model is an analytic function of crevasse depth with no real complications. Even the $F(\lambda)$ term, which is the apparent source of complication where the authors therefore inserted their observed crevasse depths, is a simple polynomial function of crevasse depth (see van der Veen 1998, Equation 6). This is easily solved for crevasse depth (especially when crevasses are shallow compared to the ice thickness), as with the other simple models used.

5. Contradictions in conclusions: The authors state that "Broadly, we see no clear, consistent patterns in either the crevasse density or depths as one moves from the glacier interior towards the glacier terminus" (line 256). This is in conflict with the manuscript's title and with other statements in the paper. For instance, they say that the models "fail to reproduce realistic along-flow variations in crevasse depth" (line 354) and refer to "the observed along-flow increase in maximum crevasse depths at over half of our glaciers" (line 398) and to "large-scale spatial patterns in crevasse depths" (line 402). I am left confused as to whether the authors find along-flow trends or patterns, or not.

6. Figure 4 shows the continuing limitations of these depth estimates. The maximum crevasse depth observed in along-flow bins approaches the modeled depth as the bins get larger (e.g., panel h). (A minor note: The caption should say which model (minimal Nye) was used.) It seems likely to me that the deepest crevasses (panel h) are also the widest crevasses, which gives the ATM a better look and increases its chance of observing the true point of the V. I believe the authors have the data to test this. If so, they could (and should) exclude crevasses whose width-to-depth ratio is smaller than what the ATM can observe (see my first review). Throwing out meaningless data would undoubtedly improve the analysis.

**Minor comments**

Lines 12–14 – This sentence says that, with a little embellishment, the observations agree with themselves.

Line 131 – This notation is unusual and informal and should be fixed.

Line 167 – The mean depth difference between scans is reported as a negative value. It should be positive, which calls into question how this difference was calculated. The given explanation for the difference between scans (changes in crevasse wall slopes) is unlikely – crevasse geometries should not change meaningfully over a few hours in plug flow regimes. I think it is more likely that this

relatively small depth difference arises from how the crevasse depths were calculated from oblique observations of the crevasse walls.

Line 275 – Strain is incorrectly defined here as the spatial integration of strain rate. This may be a typo (easily fixed) or, if the analysis was actually performed this way, a bigger flaw that would affect the results.

Lines 300, 320 – "oscillation" requires a period or some regularity, which is not seen here. A better description would be "variation" or "fluctuation".

Lines 380–381 – "Viscous" and "ductile" are not antonyms. "Brittle" is the opposite of "ductile".

Lines 383–384 – I agree that large spatial gradients in the water table are unrealistic. But the water table shouldn't follow "regional patterns in meltwater runoff", especially at the 100-m scale; instead, it should follow basal hydrology, which is variable across short length scales, even on outlet glaciers. It's neither feasible nor useful to compare the crevasse water depths to patterns in basal water pressure, but I thought I could point out this error.

Equation 1 – This should be written as $d_{modeled} = ...$ and the depth should be named $d_{LEFM}$ or some other name to distinguish it from the other $d_{modeled}$ (Nye model) in Equation 2. All equations should be moved out of the Introduction section.

Equation 2 – What values were used for $\dot{\epsilon}_{crit}$ and $A$? I believe these choices are described in the Methods, so this would be another benefit to moving equations to Methods.

Equation 5 – Use of Glen's Flow Law on crevassed ice is problematic because opening of the crevasses (which is reflected in the measured strain rates) relieves stresses in a way that the continuum-based law does not capture. (This is related to the deformation enhancement, $D$, that immediately precedes this section of the manuscript, so I'm surprised the authors didn't catch, or at least comment on, this.) Thus, the calculated stresses (Equation 5) are higher than the stresses actually sustained in the ice.

Figure S1 – The thick lines and high volume of data on each panel makes the data hard to see.

---

## Author Response (AR2)

Dear Editor and Reviewer,

Thank you for your additional comments on the manuscript. In response to the suggestions we received regarding our last draft, we have made revisions throughout the text to tighten our focus on the observed patterns of open crevasses and the relationship between these observations of strain rates. While we compare our observed crevasses to predictions from the Nye and LEFM crevassing models, we have removed our emphasis on a thorough model evaluation. Additionally, we strengthen the discussion regarding the limitations of our observations so that it is clear that the observations and models may not be correlated for a number of reasons. However, we retain our central statement that the absence of modeled crevasses in regions where deep crevasses are found (tens of meters for our observed depths, which are minimum estimates) is problematic for models. We believe that, through this most recent round of revisions, we have addressed the concerns raised by one of the reviewers and by the Editor, while simultaneously retaining the impact of our novel study and the unprecedented evaluation of crevasse patterns on Greenland glaciers.

Please do not hesitate to contact corresponding author Enderlin if you have any questions regarding our revisions.

Sincerely,
Ellyn Enderlin & Tim Bartholomaus

Reviewer comments in black, responses in blue.
* * *
Summary of manuscript

This manuscript describes a new, large dataset of lidar-derived estimates of crevasse depths on Greenland outlet glaciers and compares these depths to calculations by two simple, physically based models for crevasse depths. As in its earlier draft, the manuscript reports that the data-derived and model-based depths are "unrelated" and thus crevasse depth models have questionable utility. Compared to the earlier draft, this version briefly acknowledges the shortcomings in the central dataset, and adds comparison with a second crevasse depth model.

Concerns about validation

New datasets, particularly ones as large in scope as the one presented here, are typically validated before they are used. Unfortunately, no validation was done for this dataset. Therefore, it is hard to have confidence in the authors' conclusions that local stress models are inaccurate. This is my first major criticism with the work presented.

We appreciate the value of on-the-ground measurements of fracture geometries, if such measurements were possible. Unfortunately, there is no present method that would allow us to observe the full penetration depths for crevasses, or even a method to measure the microfractures that likely extend below open crevasses (that we know of). We now explicitly state this

limitation in the manuscript. Despite our inability to make in-situ measurements of microscopic crack geometries, our direct measurements of the surface shapes of Greenland glaciers, including open crevasses, constitute novel and unprecedented measurements themselves. As we state in the manuscript, these observed, minimum crevasse depths are valuable first-order estimates of crevassing that are needed to advance our understanding of this important component of glacier systems.

The authors' justification of their V-shape model (lines 131-138) remains problematic. They advocate that a crevasse initially forms as a V, and later processes, like serac toppling, cause deviations from a V. This manuscript assumes "negligible fracture extent beyond the bottom of the V", but as I pointed out in my previous review, crevasses in lower surface stress regimes are more irregular than a V and tend to quickly go off-nadir with depth. For example, Figure 1a in Smith et al. (2015) is a field photo that shows a near-surface cross section of a Greenland crevasse in a lower-stress setting. The surface expression (a V, rough depth estimate 3 meters based on the scale of the person in the background) gives way to a curved, inclined fracture of at least the same depth beneath the tip of that V. For the outlet glacier crevasses the authors here are studying, the deviatoric stresses are higher and the V shape extends much deeper, but below the tip of the V, where the stresses decay, this field photo should be representative. Clearly, beneath the tip of the V lies a narrow, fractured plane that is not measureable by a lidar with a 1 meter footprint.

We have made revisions throughout the manuscript to clarify that our crevasse depths represent minimum estimates since they only measure the open portion of crevasses. We acknowledge in several places that fracturing will likely extend for an unknown amount below the open portion of the crevasses.

The authors attempt to quantify the ability of the lidar to locate the "true crevasse bottom" by comparing up-glacier to down-glacier swaths (lines 155-158). Unfortunately, this does not actually evaluate the accuracy of the lidar, merely its repeatability. The 1 meter footprint of the lidar is too large to measure sub-centimeter fractures, and this will be the case regardless of the direction of flight or the time of day.

We have revised the text here to make it clear that our uncertainty estimates represent uncertainties associated with the estimates of depth for the open portion of crevasses, not the full fracture depth.

The revision contains a few added sentences (lines 85-89) that address (but does not put to rest) the above concern.
"While the full crevasse depth is unmeasurable from glacier surfaces, we assume here that the depth of the visible, near-surface void space is positively correlated with the full depth of the crevasse, which likely extends beyond the depth of the void space. We refer to the maximum, visibly open depth below the surface as the observed crevasse depth."

First, "observed crevasse depth" is misleading, since full crevasse depth is not actually being observed. I suggest "observable depth" or "observed minimum depth", which I perceive as more accurate.

We have kept "observed" crevasse depth throughout since this is the depth that can be observed from the surface (there is no method to observe the microfracture below the open portion) but have repeatedly clarified the limitations of our observations. In locations where the limitations are particularly important to recall, we add the word "minimum" when referring to our observations.

More importantly, this assumption needs to be tested (the dataset validated). It may well be the case that the depth of the V is correlated to the full depth of the crevasse. However, it may not. It is unreasonable to discredit long-standing theories with an unvalidated dataset that the authors "assume" is correlated with truth.
We have removed language that suggests the observed crevasse depths are perfectly correlated with full fracture depths.

Concerns about correlation analysis
The authors have reframed some of their discussion to emphasize the lack of correlation between
"observed" crevasse depth and modeled or expected crevasse depth. However, a lack of correlation should be the expected result, due to the two significant limitations inherent to the dataset: (1) scan angle limitations described in my first review, and (2) the false approximation of a V-shape described above. For both these reasons, and as now mentioned in the manuscript (lines 131-138), the crevasse depth "observations" are minima.
An attempt to compare truncated depth "observations" with model-based predictions that span a
full range of depths will clearly return no correlation. This is indeed what the authors find, yet they attribute this to deficiencies in the models.
The above forms my second major concern with the analysis: the known bias in the dataset renders it (as currently presented) unable to test the accuracy of any models. One idea the authors might pursue to address limitation (1) above is selecting only crevasses whose sensed depth-to-width ratios are within the range of the lidar.
We have clarified that we present minimum crevasse depths that are only valid for the open portion of crevasses throughout the manuscript. We have removed most of the quantitative comparison of observed and modeled crevasse depths and now make it more explicit that observational limitations may also influence the comparison. However, we have also emphasized that the major finding is that the local stress models predict an absence of crevasses where we observed minimum depths on the order of tens of meters.

Review conclusions
Overall, the dataset requires validation, and the manuscript needs to more responsibly represent the limitations of the "observed depth" dataset throughout the manuscript. It is evident that a lot of work went into producing this dataset. Therefore, I hope that the authors can think deeper into the limitations of the data, then validate the dataset so that it may be used for analysis.

[revised manuscript text omitted]

**Page 11: [1] Deleted**          **Revised 2020 Apr 1**          **4/1/20 4:08:00 PM**

**Page 11: [1] Deleted**          **Revised 2020 Apr 1**          **4/1/20 4:08:00 PM**

**Page 11: [1] Deleted**          **Revised 2020 Apr 1**          **4/1/20 4:08:00 PM**

**Page 11: [1] Deleted**          **Revised 2020 Apr 1**          **4/1/20 4:08:00 PM**

**Page 11: [1] Deleted**          **Revised 2020 Apr 1**          **4/1/20 4:08:00 PM**

**Page 11: [1] Deleted**          **Revised 2020 Apr 1**          **4/1/20 4:08:00 PM**

**Page 11: [1] Deleted**          **Revised 2020 Apr 1**          **4/1/20 4:08:00 PM**

**Page 11: [1] Deleted**          **Revised 2020 Apr 1**          **4/1/20 4:08:00 PM**

**Page 11: [1] Deleted**          **Revised 2020 Apr 1**          **4/1/20 4:08:00 PM**

**Page 11: [1] Deleted**          **Revised 2020 Apr 1**          **4/1/20 4:08:00 PM**

**Page 11: [1] Deleted**          **Revised 2020 Apr 1**          **4/1/20 4:08:00 PM**

**Page 11: [1] Deleted**          **Revised 2020 Apr 1**          **4/1/20 4:08:00 PM**

**Page 11: [1] Deleted**          **Revised 2020 Apr 1**          **4/1/20 4:08:00 PM**

**Page 11: [2] Deleted**          **Revised 2020 Apr 1**          **4/1/20 4:08:00 PM**

**Page 11: [2] Deleted**          **Revised 2020 Apr 1**          **4/1/20 4:08:00 PM**

**Page 11: [3] Deleted**          **Revised 2020 Apr 1**          **4/1/20 4:08:00 PM**

**Page 11: [3] Deleted** | **Revised 2020 Apr 1** | **4/1/20 4:08:00 PM**

**Page 11: [3] Deleted** | **Revised 2020 Apr 1** | **4/1/20 4:08:00 PM**

**Page 11: [3] Deleted** | **Revised 2020 Apr 1** | **4/1/20 4:08:00 PM**

**Page 11: [3] Deleted** | **Revised 2020 Apr 1** | **4/1/20 4:08:00 PM**

**Page 11: [3] Deleted** | **Revised 2020 Apr 1** | **4/1/20 4:08:00 PM**

**Page 11: [3] Deleted** | **Revised 2020 Apr 1** | **4/1/20 4:08:00 PM**

**Page 11: [4] Deleted** | **Revised 2020 Apr 1** | **4/1/20 4:08:00 PM**

**Page 11: [4] Deleted** | **Revised 2020 Apr 1** | **4/1/20 4:08:00 PM**

**Page 11: [4] Deleted** | **Revised 2020 Apr 1** | **4/1/20 4:08:00 PM**

**Page 11: [4] Deleted** | **Revised 2020 Apr 1** | **4/1/20 4:08:00 PM**

**Page 11: [4] Deleted** | **Revised 2020 Apr 1** | **4/1/20 4:08:00 PM**

**Page 11: [4] Deleted** | **Revised 2020 Apr 1** | **4/1/20 4:08:00 PM**

**Page 11: [4] Deleted** | **Revised 2020 Apr 1** | **4/1/20 4:08:00 PM**

**Page 11: [5] Deleted** | **Revised 2020 Apr 1** | **4/1/20 4:08:00 PM**

**Page 11: [5] Deleted** | **Revised 2020 Apr 1** | **4/1/20 4:08:00 PM**

**Page 11: [5] Deleted** | **Revised 2020 Apr 1** | **4/1/20 4:08:00 PM**

| Page 11: [5] Deleted | Revised 2020 Apr 1 | 4/1/20 4:08:00 PM |
|---|---|---|

| Page 11: [5] Deleted | Revised 2020 Apr 1 | 4/1/20 4:08:00 PM |
|---|---|---|

| Page 11: [5] Deleted | Revised 2020 Apr 1 | 4/1/20 4:08:00 PM |
|---|---|---|

| Page 11: [5] Deleted | Revised 2020 Apr 1 | 4/1/20 4:08:00 PM |
|---|---|---|

| Page 11: [5] Deleted | Revised 2020 Apr 1 | 4/1/20 4:08:00 PM |
|---|---|---|

| Page 11: [5] Deleted | Revised 2020 Apr 1 | 4/1/20 4:08:00 PM |
|---|---|---|

| Page 11: [5] Deleted | Revised 2020 Apr 1 | 4/1/20 4:08:00 PM |
|---|---|---|

---

## Author Response (AR3)

Dear Editor and Reviewer,

Thank you for your additional comments on the manuscript. We have rearranged portions of the text and revised wording in line with several of the reviewer's comments in order to clearly convey the major implications of our analysis. We believe that we have addressed the concerns raised by the reviewer, while simultaneously retaining the impact of our novel study and the unprecedented evaluation of crevasse patterns on Greenland glaciers.

Please do not hesitate to contact corresponding author Enderlin if you have any questions regarding our revisions.

Sincerely,

Ellyn Enderlin & Tim Bartholomaus

Reviewer comments in black, responses in blue.
* * *
Summary of manuscript

This iteration of the Greenland airborne crevasse depth analysis acknowledges the limitations to the observations. The analysis is mostly the same as presented before, with additions to the discussion and changes to the conclusions. The new conclusion – that advection and/or blunting influence crevasse depth but are not included in the usual simple models tested here – is reasonable, but strangely does not appear in the abstract. The revision still does not incorporate the known uncertainties in the crevasse depth dataset, which therefore still looks like noise to me. I suggest that the authors remove too-narrow crevasses (where the ATM cannot sense the tip of the V) from their dataset – perhaps I should have been more specific, in previous reviews, that this should have been done. This should be simple to implement using the data the authors already have.

We agree with the reviewer that the lidar may not map the base of narrow crevasses given its off-nadir geometry. We have explicitly acknowledged this limitation in the text. However, we do not attempt to filter crevasses for potential data gaps as these are not obvious in our dataset and we only report the observed depth when extrapolation of the fjord wall slopes yields a shallower-than-observed depth. As such, the lidar's inability to map the bottom of narrow but deep crevasses should not impact the patterns in observed crevasse depth reported here.

Overall, the paper is improved from its earlier versions, but suffers from new issues of clarity and organization, and lingering science problems.

Major comments

1. The description of the modeling analysis is convoluted and spread across multiple sections. Equations are presented in the Introduction and built on in the Methods. A formula for the LEFM model is never given; I think it is intertwined

with the Nye formulation, but I am not sure. A standard approach would define modeled crevasse depth $d_X$ = for each model X in the Methods. This would make it easy to compare the factors and adjustments that each model incorporates.

In line with the reviewer's suggestion, we have moved the equations from the Introduction to the Methods. We have also clarified how Eqn. 1 can be used to solve for the crevasse depth using the LEFM formulation.

2. The abstract ends with a punt: "We therefore suggest that additional analyses of the controls on crevassing are performed prior to implementation of either crevassing model within ice sheet models." It seems that the authors land on advection (line 412) as the source of the misfit between estimated and modeled crevasse depth, although they do not quantitatively explore it. Adding the advection hypothesis to the abstract, and supporting it with a back of the envelope calculation in the paper itself, would strengthen it.

We agree with the reviewer that the final sentences of the abstract could be tightened to focus on the major points of the revised manuscript. We have now been more explicit in our recommendation for further validation, acknowledging that the lidar-derived crevassed depths are not sufficient for teasing-out controls on crevassing and that in situ data are needed. We have also revised the final sentence to specifically state that advective processes must be explored in more detail to ensure that we are adequately simulating realistic crevasse patterns in models that rely on crevassing to trigger changes in dynamics.

• Advection estimate: Figure 5a, Nye Minimal or Nye Critical model, shows compressive regions up to ~1 km long. With estimated flow speeds 1 km/yr, this means that crevasses that form upstream must persist for ~1 year on their transit through compressive areas. This is an entirely reasonable lifetime for a crevasse, even on an outlet glacier. This could be supported with observations (e.g., tracking in satellite images) or cited (I am not sure of a reference) or left as a statement (which I would believe).

Surface velocities are on the order of kilometers per year for these outlet glaciers so it is easy to envision that crevasses can traverse compressional zones over the course of months. It is beyond the scope of this paper, however, to track individual crevasse evolution or to model the fate of crevasses as they move through compressional zones. Given that crevasse closure is as complicated as crevasse fracture, we do not think a simple calculation of crevasse residence time in a compressional zone will provide any additional support to test our hypothesis that advection influences crevasse distributions for fast-flowing glaciers.

3. Deformation enhancement factor: The name is misleading. The term D is presented as a reduction in viscosity due to fractures, roughly in line with Borstad et al. (2016). However, Borstad did a full L-curve analysis to balance smoothness in their D field against the misfit that D attempts to minimize. What is presented here, on the other hand, is pure misfit (Equation 4). It is disingenuous to calculate misfit and name it something else, effectively ascribing all of the misfit to a particular process. The authors do seem to acknowledge this: "such detailed tuning is neither physically motivated nor practical" (line 230) and "the optimal deformation enhancement and water depth tuning parameters found here have no physical basis" (line 387), but they continue with their analysis anyway. Unfortunately, this sort of dismissing of warning signs is a recurring pattern in the analysis.

We appreciate that we did not perform the same robust modeling analysis of Borstad et al. (2016) and we have slightly revised the text in this section to make our intention clearer. We do not attribute the misfit to damage alone ("our deformation enhancement factor is likely a function of spatial variations in damage, ice temperature, and crystal fabric") but explore whether the use of a viscosity parameter that varies along flow will improve agreement between observed and modeled crevasse depths. We have amended the sentence that the reviewer has called out, to make it clear that we do not assume that the minimization of the misfit at each crevasse can be totally attributed to uncertainty in viscosity: "However, such detailed tuning is neither physically motivated nor practical for models since numerous processes can contribute to the misfit between observed and modeled crevasse depths…". We have not removed this section from our

analysis, however, since it represents one of three misfit-minimizing modifications that we made to relatively unconstrained parameters in the Nye formulation.

4. A priori information on crevasse depth: The authors fold their crevasse depth observations into the LEFM model (line 251), which they find improves the agreement with the data. Of course it does. However, they need not (and should not) have done this, as the LEFM model is an analytic function of crevasse depth with no real complications. Even the $F(\lambda)$ term, which is the apparent source of complication where the authors therefore inserted their observed crevasse depths, is a simple polynomial function of crevasse depth (see van der Veen 1998, Equation 6). This is easily solved for crevasse depth (especially when crevasses are shallow compared to the ice thickness), as with the other simple models used.

   The reviewer is correct that we could have performed the LEFM calculations without using a priori information to constrain the F term, which accounts for the influence of crevasse shape on the stress intensity factor. However, we had used our a priori information on crevasse sizes to minimize the observed-Nye misfit and our goal here was to also use that information to minimize the observed-LEFM misfit in order to robustly compare models when informed with a priori data. We have now made it clearer that we used Eqn. (6) in van der Veen (1998) to calculate our lambda, which was used to solve for F.

5. Contradictions in conclusions: The authors state that "Broadly, we see no clear, consistent patterns in either the crevasse density or depths as one moves from the glacier interior towards the glacier terminus" (line 256). This is in conflict with the manuscript's title and with other statements in the paper. For instance, they say that the models "fail to reproduce realistic along-flow variations in crevasse depth" (line 354) and refer to "the observed along-flow increase in maximum crevasse depths at over half of our glaciers" (line 398) and to "large-scale spatial patterns in crevasse depths" (line 402). I am left confused as to whether the authors find along-flow trends or patterns, or not.

   We have revised the first statement that the authors mention here to make it clearer that we mean there is no clear, simple pattern across all glaciers in our analysis, not that the crevasses are totally randomly or uniformly distributed.

6. Figure 4 shows the continuing limitations of these depth estimates. The maximum crevasse depth observed in along-flow bins approaches the modeled depth as the bins get larger (e.g., panel h). (A minor note: The caption should say which model (minimal Nye) was used.) It seems likely to me that the deepest crevasses (panel h) are also the widest crevasses, which gives the ATM a better look and increases its chance of observing the true point of the V. I believe the authors have the data to test this. If so, they could (and should) exclude crevasses whose width-to-depth ratio is smaller than what the ATM can observe (see my first review). Throwing out meaningless data would undoubtedly improve the analysis.

   We have revised the caption to make it clearer that these are the modeled depths using the minimal Nye model. It is important to note that we report the extrapolated depths when deeper than the observed depths. In many cases, the widest crevasses are littered with serac topple debris or even water (based on inspection of satellite images), so they are no more likely to have their true open depth mapped than the crevasses that are narrower. The improved agreement in observed and modeled maximum crevasse depths in Figure 4 is largely due to the spatial averaging of strain rates over progressively larger scales, which minimizes the prevalence of compressional zones and reduces the magnitude of extensional strain rates.

Minor comments

Lines 12–14 – This sentence says that, with a little embellishment, the observations agree with themselves.

We agree but have not removed this sentence because it is important to note that a prior information of crevasse depth from lidar, such as presented here, can be used to reasonably simulate large-scale variations in crevasse depth using the LEFM model.

Line 131 – This notation is unusual and informal and should be fixed.

Revised.

Line 167 – The mean depth difference between scans is reported as a negative value. It should be positive, which calls into question how this difference was calculated. The given explanation for the difference between scans (changes in crevasse wall slopes) is unlikely – crevasse geometries should not change meaningfully over a few hours in plug flow regimes. I think it is more likely that this relatively small depth difference arises from how the crevasse depths were calculated from oblique observations of the crevasse walls.

We have revised this sentence to make it clearer that we were referring to the effect of the imaging angle on the calculated wall slopes. This difference was calculated by subtracting depths from repeat scans, as described in the text, resulting in either a positive or negative number depending on whether the repeat depth was shallower or deeper than the initial depth.

Line 275 – Strain is incorrectly defined here as the spatial integration of strain rate. This may be a typo (easily fixed) or, if the analysis was actually performed this way, a bigger flaw that would affect the results.

We have changed this definition to "time-integrated longitudinal strain rate" to better reflect the calculation.

Lines 300, 320 – "oscillation" requires a period or some regularity, which is not seen here. A better description would be "variation" or "fluctuation".

Replaced with "fluctuation"

Lines 380–381 – "Viscous" and "ductile" are not antonyms. "Brittle" is the opposite of "ductile".

Removed

Lines 383–384 – I agree that large spatial gradients in the water table are unrealistic. But the water table shouldn't follow "regional patterns in meltwater runoff", especially at the 100-m scale; instead, it should follow basal hydrology, which is variable across short length scales, even on outlet glaciers. It's neither feasible nor useful to compare the crevasse water depths to patterns in basal water pressure, but I thought I could point out this error.

Removed the mention of runoff variability

Equation 1 – This should be written as $d_{modeled} = ...$ and the depth should be named $d_{LEFM}$ or some other name to distinguish it from the other $d_{modeled}$ (Nye model) in Equation 2. All equations should be moved out of the Introduction section.

Revised

Equation 2 – What values were used for $\dot{\varepsilon}_{crit}$ and A? I believe these choices are described in the Methods, so this would be another benefit to moving equations to Methods.

Clearer now that the equations have been moved to the methods and the site-specific information is provided in subsequent paragraphs.

Equation 5 – Use of Glen's Flow Law on crevassed ice is problematic because opening of the crevasses (which is reflected in the measured strain rates) relieves stresses in a way that the continuum-based law does not capture. (This is related to the deformation enhancement, D, that immediately precedes this section of the manuscript, so I'm surprised the authors didn't catch, or at least comment on, this.) Thus, the calculated stresses (Equation 5) are higher than the stresses actually sustained in the ice.

Now noted in the text

Figure S1 – The thick lines and high volume of data on each panel makes the data hard to see.

It is unclear which figure this refers to since Fig S1 does not contain thick lines.

**Sharp contrasts in observed and modeled crevasse patterns at Greenland's marine terminating glaciers**

Ellyn M. Enderlin[1], Timothy C. Bartholomaus[2]

[1]Department of Geosciences, Boise State University, Boise, Idaho, 83725, USA
[2]Department of Geological Sciences, University of Idaho, Moscow, Idaho, 83844, USA

*Correspondence to*: Ellyn M. Enderlin (ellynenderlin@boisestate.edu)

**Abstract.** Crevasses are affected by and affect both stresses and surface mass balance of glaciers. These effects are brought on through potentially important controls on meltwater routing, glacier viscosity, and iceberg calving, yet there are few direct observations of crevasse sizes and locations to inform our understanding of these interactions. Here we extract depth estimates for the visible portion of crevasses from high-resolution surface elevation observations for 52,644 crevasses from 19 Greenland glaciers. We then compare our observed depths with those calculated using two popular models that assume crevasse depths are functions of local stresses: the Nye and linear elastic fracture mechanics (LEFM) formulations. We find that neither formulation accurately captures sub-kilometer variations in observed crevasse depths. However, when informed by observed crevasse depths, the LEFM formulation produces kilometer-scale variations in crevasse depth in decent agreement with observations. Importantly, we find that along-flow patterns in crevasse depths are unrelated to along-flow patterns in strain rates (and therefore stresses), but that cumulative strain rate is moderately more predictive of crevasse depths at the majority of glaciers. While it is conceivable that our observed, lidar-derived crevasse depths have no relation to the full depth of fractures that vertically penetrate glacier ice, our findings support a need for additional testing and in situ validation of the Nye and LEFM models. Through analyses focused on reconciling crevasse distributions in extensional and compressional regions, a better understanding of crevasse advection and temporal evolution will enable more confident projection of fracture-driven terminus position change and meltwater routing.

[revised manuscript text omitted]